# Make LoRA Great Again: Boosting LoRA with Adaptive Singular Values and Mixture-of-Experts Optimization Alignment

**Chenghao Fan** [*1]   **Zhenyi Lu** [*1]   **Sichen Liu** [1]   **Chengfeng Gu** [2]   **Xiaoye Qu** [1]   **Wei Wei** [1]   **Yu Cheng** [3]

## Abstract

While Low-Rank Adaptation (LoRA) enables parameter-efficient fine-tuning for Large Language Models (LLMs), its performance often falls short of Full Fine-Tuning (Full FT). Current methods optimize LoRA by initializing with static singular value decomposition (SVD) subsets, leading to suboptimal leveraging of pre-trained knowledge. Another path for improving LoRA is incorporating a Mixture-of-Experts (MoE) architecture. However, weight misalignment and complex gradient dynamics make it challenging to adopt SVD prior to the LoRA MoE architecture. To mitigate these issues, we propose Great LoRA Mixture-of-Expert (GOAT), a framework that (1) adaptively integrates relevant priors using an SVD-structured MoE, and (2) aligns optimization with full fine-tuned MoE by deriving a theoretical scaling factor. We demonstrate that proper scaling, without modifying the architecture or training algorithms, boosts LoRA MoE's efficiency and performance. Experiments across 25 datasets, including natural language understanding, commonsense reasoning, image classification, and natural language generation, demonstrate GOAT's state-of-the-art performance, closing the gap with Full FT. Our code is available at: https://github.com/Facico/GOAT-PEFT.

## 1. Introduction

Recent large language models (LLMs) have shown impressive capabilities (Dai et al., 2024; Touvron et al., 2023; Yang et al., 2024; OpenAI et al., 2024), but fine-tuning them for downstream tasks is computationally expensive (Hu et al., 2021; Zhao et al., 2024). To reduce costs, parameter-efficient fine-tuning (PEFT) techniques (Hu et al., 2021; Pfeiffer et al., 2021; Houlsby et al., 2019; Tian et al., 2024; Fan et al., 2024b) have been proposed. Among them, LoRA (Hu et al., 2021) is popular for its simplicity and effectiveness. It reparameterizes the weight matrix $W \in \mathbb{R}^{m \times n}$ into $W = W_0 + BA$, where $W_0 \in \mathbb{R}^{m \times n}$ is a frozen full-rank matrix, and $B \in \mathbb{R}^{m \times r}, A \in \mathbb{R}^{r \times n}$ are low-rank adapters to be learned. Since the rank $r \ll \min(m, n)$, LoRA only updates a small fraction of the parameters, greatly reducing memory usage.

Despite its computational efficiency, LoRA often underperforms full fine-tuning (Full FT) (Wang et al., 2024d;c; Fan et al., 2024a), even with Mixture-of-Experts (MoE) architectures (Zadouri et al., 2024; Liu & Luo, 2024; Tian et al., 2024). Our rigorous analysis identifies two key factors limiting LoRA's performance: (1) *Suboptimal Initialization*: The isotropic random initialization for matrix $A$ and zero initialization for matrix $B$ provide a non-informative prior, resulting in unguided optimization subspaces. While Wang et al. (2024b); Meng et al. (2024) applied singular value decomposition (SVD) for better initialization, their reliance on a static, predefined subset of pre-trained weights limits the capture of the full range of pre-trained knowledge. It raises the question: *Can we adaptively integrate relevant priors of pre-trained knowledge based on input?* (2) *Unaligned Optimization*: Furthermore, the intrinsic low-rank property of LoRA leads to large gradient gaps and slow convergence in optimization, therefore underperforming Full FT. In LoRA MoE scenarios, the total rank is split among experts, resulting in lower ranks and further increasing this challenge. Existing strategies (Wang et al., 2024c;d) focus only on single LoRA architectures and ignore the added complexity of random top-$k$ routing and multiple expert weights within MoE architecture. When SVD-based initialization is applied to LoRA MoE, weight alignment becomes a challenge, which has never been considered in previous methods that used zero initialization. This further raises the question: *How do we mitigate the optimization gap in LoRA MoE initialized with prior information?*

To address these challenges, we propose **GOAT** (Great LoRA Mixture-of-Experts), which employs an SVD-

---

[*]Equal contribution  [1]School of Computer Science & Technology, Huazhong University of Science and Technology [2]Zhejiang University [3]The Chinese University of Hong Kong. Correspondence to: Wei Wei <weiw@hust.edu.cn>.

*Proceedings of the 42nd International Conference on Machine Learning*, Vancouver, Canada. PMLR 267, 2025. Copyright 2025 by the author(s).

structured MoE with theoretical scaling to match full fine-tuning performance. Our method highlights two important innovations. (1) *Initialization*: We demonstrate that different segments of pre-trained knowledge in the SVD structure are crucial depending on the input. To capture this adaptively, we propose initializing LoRA MoE experts with distinct singular value segments, with the router selecting the appropriate prior information. (2) *Optimization*: Rather than directly targeting the gap with full fine-tuning, we focus on an upcycled MoE [2] with full-rank fine-tuning. We show that when each low-rank expert plus pre-trained weight approximates its full-rank counterpart, the router's behavior remains consistent, enabling effective optimization of expert weights. Through simple scaling, without altering architecture or algorithms, we significantly improve both convergence speed and performance. We also derive the optimal weight alignment strategy and a theoretical scaling scheme for better gradient alignment.

In summary, the contributions of our method are as follows:

- *Adaptive Priors Initialization*: We propose a novel SVD-structured MoE framework that adaptively integrates pre-trained knowledge, addressing the limitations of non-informative or static priors.
- *Theoretical Optimization Alignment*: We reveal a key connection between LoRA and full fine-tuning upcycled MoE, deriving an optimal weight alignment strategy and scaling scheme to close the performance gap.
- *State-of-the-Art Performance*: Extensive experiments on 25 tasks demonstrate that our method achieves superior performance while maintaining scalability.

## 2. Background and Motivation

### 2.1. Rethinking Singular-Value Initialization

Singular-value initialization is widely used in LoRA to preserve pre-trained weight characteristics (Zhao et al., 2024; Meng et al., 2024; Wang et al., 2024a; Lu et al., 2024). PiSSA (Meng et al., 2024) only updates the largest singular values, while MiLoRA (Wang et al., 2024b) adjusts minor singular values for strong performance.

To unify SVD-based methods with full fine-tuning, let $W_0 \in \mathbb{R}^{m \times n}$ be the pre-trained weight with SVD, $W_0 = U\Sigma V^\top$. Assuming $h = \min(m, n)$ and LoRA rank $r$, we decompose $W_0$ into rank-$r$ blocks:

$$W_0 = \sum_{i=0}^{l} U_i \Sigma_i V_i^\top, \qquad (1)$$

where $l = \frac{h}{r} - 1$ and $i$ denotes the segment $[i \cdot r : (i+1) \cdot r]$. The submatrices are defined as $U_i = U_{[i \cdot r:(i+1)\cdot r,:]} \in$

---

[2] Upcycled MoE initializes all experts with the same pre-trained weights, which we adopt for simplicity.

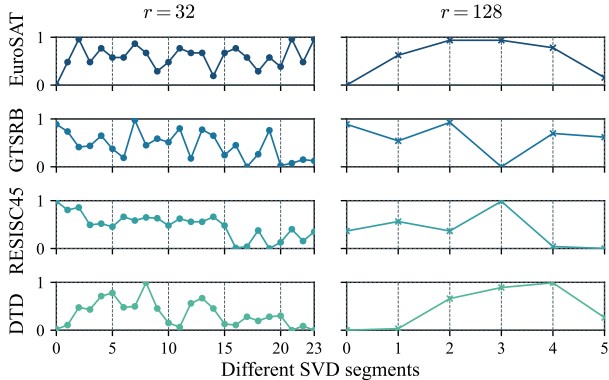

*Figure 1.* The effect of initializations from different SVD segments $(u_i, \sigma_i, v_i^\top)$ for rank 32 and 128. The performance normalized by min-max scaling.

$\mathbb{R}^{r \times m}$, $\Sigma_i = \Sigma_{[i \cdot r:(i+1)\cdot r, i \cdot r:(i+1)\cdot r]} \in \mathbb{R}^{r \times r}$, and $V_i = V_{[i \cdot r:(i+1)\cdot r,:]} \in \mathbb{R}^{r \times n}$. Fine-tuning methods are represented as:

$$
\begin{aligned}
\text{Full FT}: \quad & U_0\Sigma_0 V_0^\top + U_1\Sigma_1 V_1^\top + \cdots + U_l\Sigma_l V_l^\top \\
\text{PiSSA}: \quad & U_0\Sigma_0 V_0^\top + (U_1\Sigma_1 V_1^\top + \cdots + U_l\Sigma_l V_l^\top)^* \\
\text{MiLoRA}: \quad & (U_0\Sigma_0 V_0^\top + \cdots + U_{l-1}\Sigma_{l-1} V_{l-1}^\top)^* + U_l\Sigma_l V_l^\top \\
\text{KaSA}: \quad & (U_0\Sigma_0 V_0^\top + \cdots + U_{l-1}\Sigma_{l-1} V_{l-1}^\top)^* + U^{\text{r}}\Sigma^{\text{r}} V^{\text{r}\top}
\end{aligned}
$$
$$(2)$$

Here, $(\cdot)^*$ denotes frozen components, while non-frozen components initialize LoRA:

$$B = U_i\Sigma_i^{1/2} \in \mathbb{R}^{m \times r}, \quad A = \Sigma_i^{1/2} V_i^\top \in \mathbb{R}^{r \times n}. \qquad (3)$$

We observe PiSSA freezes minor singular values and fine-tunes only the components $U_0\Sigma_0 V_0^\top$ with the largest norms, achieving the optimal approximation to $W_0$.[3] In contrast, MiLoRA and KaSA retain segment $0 \sim (l-1)$ as preserved pretrained knowledge, but KaSA treats the minor $U_l\Sigma_l V_l^\top$ as noise and replaces it with a new random $U^{\text{r}}\Sigma^{\text{r}} V^{\text{r}\top}$. In practice, PiSSA converges faster by focusing on principal singular values, while MiLoRA and KaSA preserve more pre-trained knowledge for better final performance.

This raises the question: *Is it reasonable to use only the principal or minor part as a fine-tuning prior?* Figure 1 illustrates the performance of fine-tuning from different segments $(U_i, \Sigma_i, V_i^\top), i \in [0, \cdots, l]$, where each segment is used for initialization while others remain frozen. The x-axis represents segment indices (*e.g.*, $x = 0$ for PiSSA, $x = l$ for MiLoRA), and the y-axis shows min-max normalized performance. We can identify two notable observations: (1) *The same initialization exhibits varying trends for different datasets*. For example, $x = l$ achieves better results on

---

[3] Proof in Appendix B

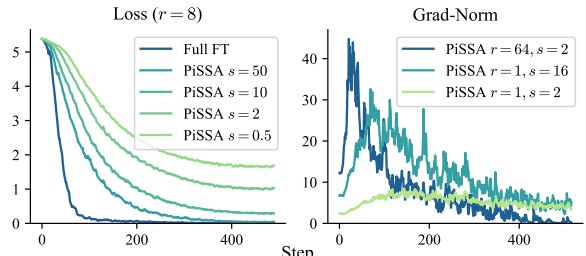

*Figure 2.* SVD initialization *vs.* scaling $s$ and rank $r$

the EuroSAT dataset, while $x = 0$ performs better on the GTSRB dataset. (2) *Middle segments play a crucial role. e.g.*, when $r = 128$, the highest performance is typically observed in the middle segments. These findings suggest that each singular value segment contains task-specific information, motivating us to allow the model to automatically select segments during optimization, leveraging all singular values while preserving the original pre-trained matrix characteristics.

### 2.2. Rethinking Scaling Factor

In LoRA, it is common practice to use the scaled variant $W = W_0 + sBA$, yet the effects of scaling factor $s$ have not been fully explored. Biderman et al. (2024) consider $s$ should typically set to 2. The SVD-based method (Meng et al., 2024) empirically makes $sBA$ independent of $s$ by dividing $B$ and $A$ by $\sqrt{\frac{1}{s}}$, while Tian et al. (2024) use larger scaling for LoRA MoE to achieve better performance.

To investigate it, as illustrated on the left of Figure 2, we first adjust $s$ in the SVD-based LoRA with a fixed rank, revealing that $s$ still impacts the convergence speed. To study the effect, we introduce the equivalent weight and gradient to quantify the gap between LoRA and Full FT.

**Definition 2.1** (Equivalent Weight and Gradient). For LoRA optimization, we define the equivalent weight as:

$$\tilde{W} \triangleq W + sBA, \tag{4}$$

The equivalent gradient of $\tilde{W}$ is defined as:

$$\tilde{g} \triangleq \frac{\partial L}{\partial \tilde{W}} \tag{5}$$

where $s$ is the scaling factor, and $G^A$ and $G^B$ are gradients with respect to $A$ and $B$, respectively.

**Lemma 2.2.** *Let $g_t$ be the gradient in full-tuning, and $B$, $A$ be the low-rank weights. At the $t$-th optimization step, the equivalent gradient can be expressed as:*

$$\tilde{g}_t = s^2 \left( B_t B_t^\top g_t + g_t A_t^\top A_t \right) \tag{6}$$

The formula for SVD-based initialization is:

$$\tilde{W} \propto sBA = s \left( \frac{1}{s} U_r \Sigma_r V_r^T \right) = U_r \Sigma_r V_r^T \tag{7}$$

$$\tilde{g} = s^2 \left( \frac{1}{s} U_r U_r^\top g + \frac{1}{s} g V_r V_r^T \right) = s \left( U_r U_r^\top g + g V_r V_r^T \right) \tag{8}$$

Though the equivalent weight is independent of $s$, equivalent gradient is proportional to $s$. As shown in Figure 2, $s = 2$ is too small. Increasing the scaling factor in SVD-based methods boosts the gradient, leading to faster convergence.

Next, we examine the effect of different ranks, as shown in Figure 2. With low rank (*e.g.*, $r = 1$), the gradient norm is small and deviates from the trend of $r = 64$, creating a performance gap (95.77 vs. 98.55). However, applying proper scaling ($s = 16$) increases the gradient norm, reducing the performance gap (from 95.77 to 97.70). This is especially beneficial in MoE scenarios, where the total rank is split among experts, resulting in lower ranks. Increased scaling can compensate for this, as supported by Tian et al. (2024).

## 3. Method

### 3.1. LoRA MoE Architecture

**Mixture-of-Experts (MoE)** An MoE layer (Qu et al., 2024; Zhu et al., 2024a;b; Zhang et al., 2024) comprises $E$ linear modules $\{W_1, \ldots, W_E\}$ and a router $W_z \in \mathbb{R}^{m \times E}$ that assigns input $\mathbf{x}$ to experts based on routing scores:

$$p^i(\mathbf{x}) = \frac{\exp(z^i(\mathbf{x}))}{\sum_{j=1}^{E} \exp(z^j(\mathbf{x}))}, \tag{9}$$

where $z(\mathbf{x}) = W_z \mathbf{x}$ and $p^i(\mathbf{x})$ is the score for expert $i$.

Let $\Omega_k(\mathbf{x})$ denote the indices of the top-$k$ scores, ensuring $|\Omega_k(\mathbf{x})| = k$ and $z^i(\mathbf{x}) > z^j(\mathbf{x})$ for all $i \in \Omega_k(\mathbf{x})$ and $j \notin \Omega_k(\mathbf{x})$. Define the weights as:

$$w^i(\mathbf{x}) = \begin{cases} \frac{\exp(z^i(\mathbf{x}))}{\sum_{j \in \Omega_k(\mathbf{x})} \exp(z^j(\mathbf{x}))}, & \text{if } i \in \Omega_k(\mathbf{x}), \\ 0, & \text{otherwise.} \end{cases} \tag{10}$$

The MoE layer output is the weighted sum of the top-$k$ experts' outputs:

$$\text{MoE}(\mathbf{x}) = \sum_{i=1}^{E} w^i(\mathbf{x}) W^i(\mathbf{x}). \tag{11}$$

**LoRA MoE.** We integrate LoRA into the MoE framework, retaining the router (Equation (10)) and using the balance loss from vanilla MoE[4]. Each expert $W^i$ is replaced by low-rank matrices $B^i \in \mathbb{R}^{m \times d}$ and $A^i \in \mathbb{R}^{d \times n}$, where $d = \frac{r}{E}$:

$$\text{MoE}_{\text{LoRA}}(\mathbf{x}) = W(\mathbf{x}) + \sum_{i=1}^{E} w^i(\mathbf{x}) \left( sB^i A^i(\mathbf{x}) \right) \tag{12}$$

---

[4]See Appendix C

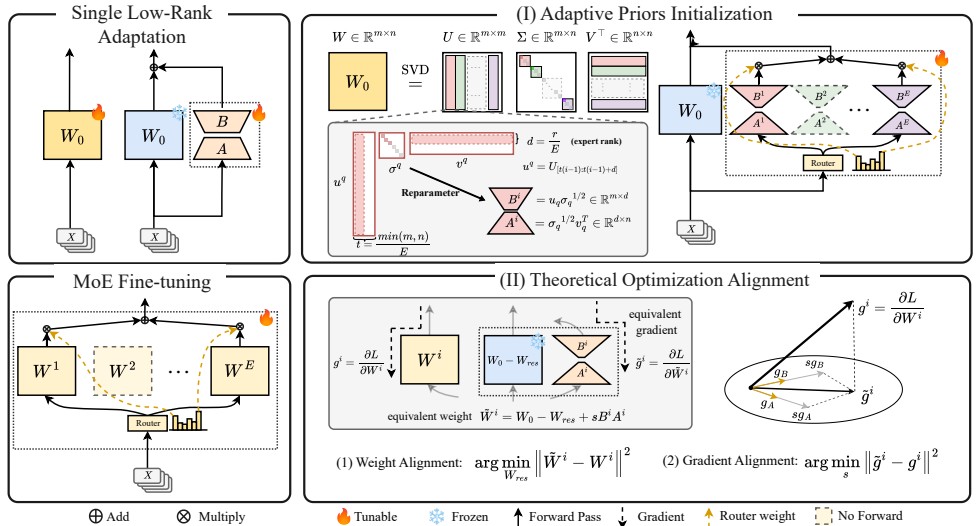

*Figure 3.* **Illustration of Our Method.** *Single Low-Rank Adaptation*: LoRA reduces trainable parameters by reparameterizing $W$ as $W = W_0 + sBA$, with $B$ and $A$ as low-rank matrices. *MoE Fine-tuning*: Full MoE fine-tuning, where experts $W^1$ and $W^E$ are selected by the router in this moment. **Subfigure (I)**: Our method replaces the single pair $B, A$ with multiple pairs $\{B^i, A^i\}_{i=1}^E$, initialized from different segments of the SVD of $W_0$ and adaptively selected by the router. **Subfigure (II)**: We align optimization with SVD-structured MoE by separately aligning each expert. $W_{\text{res}}$ ensures the equivalent weight equals $W_0$ before optimization, and we scale each expert's equivalent gradient to closely approximate full MoE fine-tuning.

where $W$ is the pre-trained weight matrix and $s$ is the LoRA scaling factor. Since $k \ll E$, LoRA MoE uses fewer active parameters than dense MoE.

### 3.2. Adaptive Priors Initialization

According to Section 2.1, the utilization of different SVD segments depends on the input. We propose initializing each expert in LoRA MoE with different SVD segments, leveraging the MoE architecture to dynamically activate experts associated with different singular values. Specially, we init expert evenly by define the set $\mathcal{E}_r$ as:

$$\mathcal{E}_r = \left\{ (U_{[:,k:k+d]}, \Sigma_{[k:k+d,k:k+d]}, V_{[k:k+d,:]}^\top) \mid j = 1, \ldots, E \right\},$$
(13)

where $t = \frac{\min(m,n)}{E}$, $k = (j-1)t$ is the starting index from segment for $j$-th expert, $d = \frac{r}{E}$ is each expert rank. Then we can construct each expert by $(U', \Sigma', V'^\top) \in \mathcal{E}_r$:

$$B_0^i = \sqrt{\frac{1}{s}} U' \Sigma'^{1/2} \in \mathbb{R}^{m \times d}, A_0^i = \sqrt{\frac{1}{s}} \Sigma'^{1/2} V'^\top \in \mathbb{R}^{d \times n}$$
(14)

The $B, A$ divide $\sqrt{s}$ to make sure that $sBA$ is independent of $s$ (Meng et al., 2024). This allows the model to adapt flexibly to various fine-tuning scenarios.

### 3.3. Theoretical Optimization Alignment

Directly applying SVD priors in MoE architectures causes weight misalignment and complex gradient dynamics, a challenge not encountered with previous zero initialization methods. Moreover, the gap in MoE-based architectures remains under-explored. We derive the following theorems

to mitigate this and show how scaling resolves the issue.

**Theorem 3.1.** *By ensuring equivalent weight $\tilde{W}_0 \approx W_0$ at initialization and maintaining equivalent gradient $\tilde{g}_t \approx g_t$ throughout optimization, we can align LoRA with Full FT. (See Definition 2.1 for equivalent weight and gradient.)*

Theorem 3.1 mitigates the performance gap in single LoRA architectures (Wang et al., 2024c;d). For MoE architectures, however, routers and top-$k$ selection complicate direct alignment. Thus, we focus on Full FT MoE and establish:

**Theorem 3.2.** *For all $i \in [1, \ldots, E]$, by ensuring equivalent weight $\tilde{W}_0^i \approx W_0^i$ at initialization and gradient $\tilde{g}_t^i \approx g_t^i$ for each expert, we can align LoRA MoE with an Upcycled MoE with full-rank fine-tuning.*

Theorem 3.2 reveals a key connection between LoRA and full fine-tuning in MoE, simplifying the problem to optimizing each expert separately. We outline the steps below.

**Initialization Alignment.** At initialization, we align the equivalent weight at initialization with an upcycled MoE, where each expert weight $\{W^i\}_{i=1}^E$ is derived from the pre-trained model's weight $W_0$ (He et al., 2024). This is equivalent to aligning $\tilde{W}_0 = W_0 + \sum_{i=1}^E w^i(\mathbf{x}) B_0^i A_0^i$ with the original weight $W_0$. As $B_0^i, A_0^i$ are initialized with prior information, We need additionally subtracting a constant $W_{\text{res}}$, ensuring the weight alignment:

$$\tilde{W}_0 = W_0 - W_{\text{res}} + \sum_{i=1}^E w^i(\mathbf{x}) s B_0^i A_0^i \approx W_0 \quad (15)$$

**Lemma 3.3.** *For all $i, j \in [1, \ldots, E]$ $(i \neq j)$:*

$$\mathbb{E}_{\mathbf{x}}[w^i(\mathbf{x})] = \frac{1}{E}, \tag{16}$$

$$Var(w^i(\mathbf{x})) = \frac{E - k}{kE^2} \tag{17}$$

**Theorem 3.4.** *Consider the optimization problem:*

$$W_{res}^+ = \arg\min_{W_{res}} \mathbb{E}_{\mathbf{x}} \left[ \left\| W_{res} - s \sum_{i=1}^{E} w^i(\mathbf{x}) B_0^i A_0^i \right\|^2 \right]. \tag{18}$$

*The closed-form solution is $W_{res}^+ = \frac{s}{E} \sum_{i=1}^{E} B_0^i A_0^i$.*

Theorem 3.4 provides an appropriate initialization scheme for MoE scenarios. Note that the original LoRA-MoE (Zadouri et al., 2024; Tian et al., 2024) uses a zero initialization scheme thus $W_{res}^+ = 0$, a special case of Theorem 3.4.

Obviously, the variance of $W_{res}^+ - s \sum_{i=1}^{E} w^i(\mathbf{x}) B_0^i A_0^i$ is proportional to $\sum_{i=1}^{E} B_0^i A_0^i$. Additionally, from Lemma 3.3, when $k$ is small (e.g., $2k < E$), $std(w^i(\mathbf{x})) > \mathbb{E}[w^i(\mathbf{x})]$. To preserve the informative SVD prior while reducing initialization instability, we scale $B_0^i A_0^i$ by $\frac{1}{\rho}$, a straightforward method to decrease variance and make a more accurate approximation in Equation (15):

$$B_0^i = \sqrt{\frac{1}{s\rho}} U_i \Sigma_i^{1/2}, A_0^i = \sqrt{\frac{1}{s\rho}} \Sigma_i^{1/2} V_i^\top \tag{19}$$

**Gradient Alignment** First, we provide the optimal scaling for zero-initialized LoRA MoE:

**Theorem 3.5.** *For $B_0 = 0, A_0 \sim U\left(-\sqrt{\frac{6}{n}}, \sqrt{\frac{6}{n}}\right), \tilde{g}_t^i = s^2 \left( B_t^i B_t^{i\top} g_t^i + g_t^i A_t^{i\top} A_t^i \right)$, and learning rate ratio between full tuning vs. LoRA is $\eta$.*

$$\arg\min_s \left\| \tilde{g}_t^i - g_t^i \right\|, \quad \forall i \in [1, \ldots, E] \tag{20}$$

*The closed-form solution of optimal scaling is $s = \sqrt{\frac{3n\eta}{r}}$.*

As $n \gg r$, it is typically the case that $s > 2$, which explains why standard scaling is insufficient and why simple scaling enhances effectiveness, as demonstrated in Section 2.2. While it is tricky to directly analyze complex gradient dynamics with SVD priors, an alternative approach is recognizing that larger scaling $s$ and $\rho$ ensure $B$ and $A$ become small and approach zero (Equation (19)), aligning with the settings in Theorem 3.5. Thus, we adopt this scaling factor in GOAT, and in practice, this approximation performs well (see Section 4.3). For scenarios with proper scaling, we extend the method to "GOAT-s", as detailed in Appendix E.

# 4. Experiment

## 4.1. Baselines

We compare GOAT with Full FT, single-LoRA, and LoRA MoE methods to substantiate its efficacy and robustness:

1. Full-Finetuning: **Full FT** fine-tunes all parameters, while **Full FT MoE** is Upcycled MoE with full-rank fine-tuning and 2 active experts out of 8 total experts.

2. Single-LoRA baselines: **LoRA** (Hu et al., 2021); **DoRA** (Liu et al., 2024); **PiSSA** (Meng et al., 2024); **MiLoRA** (Wang et al., 2024b); **rsLoRA** (Kalajdzievski, 2023); **LoRA-Dash** (Si et al., 2024); **NEAT** (Zhong et al., 2024); **KaSA** (Wang et al., 2024a)

3. LoRA MoE baselines: **MoLoRA** (Zadouri et al., 2024); **AdaMoLE** (Liu & Luo, 2024); **HydraLoRA** (Tian et al., 2024).

For a fair comparison, we closely follow the configurations from prior studies (Hu et al., 2021; Meng et al., 2024; Wang et al., 2024d). Details on the baselines are in Appendix G.

## 4.2. Datasets

We evaluate GOAT across 25 tasks, spanning 4 domains:

1. **Image Classification (IC):** We fine-tune and evaluate ViT-B/32 (Radford et al., 2021) on 7 image classification datasets (Ilharco et al., 2023).

2. **Natural Language Generation (NLG):** We fine-tune LLaMA2-7B (Touvron et al., 2023) on subset of WizardLM (Xu et al., 2023), MetaMathQA (Yu et al.) and Code-Feedback (Zheng et al., 2024). We evaluate its performance on dialogue (Zheng et al., 2023), math (Cobbe et al., 2021) and coding (Chen et al., 2021) following Wang et al. (2024d)

3. **Commonsense Reasoning (CR):** We fine-tune LLaMA2-7B on Commonsense170K and evaluate on 8 commonsense reasoning datasets (Hu et al., 2023) (multi-domain setting).

4. **Natural Language Understanding (NLU):** We RoBERTa-large (Liu et al., 2020) on 7 GLUE tasks (Wang et al., 2019) following (Hu et al., 2021).

Due to the huge memory requirements of Full FT MoE, we only evaluate it on IC and NLU tasks. Detailed of the datasets can be found in Appendix G.1.

## 4.3. Main Results

Tables 1, 2, 3 and 4 present results on 4 domain benchmarks:

- **IC** (Table 1): GOAT achieves 99.07% of full FT performance and surpasses LoRA with quadruple the parameters (rank 32). It improves 6.0% over PiSSA and 2.4% over HydraLoRA, outperforming all LoRA variants.

- **NLG** (Table 2): Our method shows the smallest performance gap with Full FT, outperforming MoLoRA by 0.25 on MTBench, 6.30% on GSM8K, and 3.14% on HumanEval, highlighting GOAT's superiority.

- **CR** (Table 3): GOAT consistently outperforms all established baselines, exceeding the best single LoRA method,

*Table 1.* We evaluate CLIP ViT-B/32 with full fine-tuning and LoRA variants with total rank 8 across StanfordCars, DTD, EuroSAT, GTSRB, RESISC45, SUN397, and SVHN datasets. **Bold** indicates the highest results.

| Method | # Params (%) | Cars | DTD | EuroSAT | GTSRB | RESISC45 | SUN397 | SVHN | Average |
|---|---|---|---|---|---|---|---|---|---|
| **Full FT** | 100 | 60.33 | 73.88 | 98.96 | 98.30 | 93.65 | 53.84 | 96.78 | 82.25 |
| **Full FT MoE** | 770 | 66.39 | 75.53 | 98.59 | 98.50 | 94.38 | 60.34 | 97.09 | 84.40 |
| *Single LoRA Methods* | | | | | | | | | |
| **LoRA** | 1.49 | 41.02 | 70.15 | 98.66 | 96.51 | 90.38 | 47.51 | 95.39 | 77.09 |
| **LoRA (rank16)** | 2.99 | 46.51 | 72.07 | 98.74 | 98.04 | 92.08 | 51.63 | 96.00 | 79.30 |
| **LoRA (rank32)** | 5.98 | 50.13 | 72.87 | 98.88 | 98.13 | 92.87 | 53.65 | 96.55 | 80.44 |
| **DoRA** | 1.49 | 40.75 | 71.91 | **98.89** | 97.71 | 90.19 | 47.54 | 95.46 | 77.49 |
| **PiSSA** | 1.49 | 40.41 | 69.62 | 98.48 | 95.84 | 90.58 | 47.21 | 95.84 | 76.85 |
| **MiLoRA** | 1.49 | 39.77 | 70.48 | 98.19 | 97.52 | 89.92 | 45.38 | 95.49 | 76.68 |
| *LoRA MoE Methods* | | | | | | | | | |
| **MoLoRA** | 2.24 | 50.83 | 73.51 | 98.63 | 97.72 | 92.58 | 52.55 | 96.00 | 80.26 |
| **AdaMoLE** | 2.33 | 49.47 | 71.65 | 98.52 | 97.73 | 91.95 | 52.29 | 95.82 | 79.63 |
| **HydraLoRA** | 1.58 | 48.42 | 72.18 | 98.40 | 97.28 | 92.93 | 51.80 | 96.06 | 79.58 |
| **GOAT** | 2.24 | **53.50** | **75.32** | 98.82 | **98.17** | 93.46 | 54.53 | 96.62 | **81.49** |

*Table 2.* We evaluate Llama-2-7B on MT-Bench, GSM8K, and HumanEval for dialogue, math, and coding.

| Method | MT-Bench | GSM8K | HumanEval |
|---|---|---|---|
| **Full FT** | 5.56 | 59.36 | 35.31 |
| *Single LoRA Methods* | | | |
| **LoRA** | 5.61 | 52.84 | 21.34 |
| **DoRA** | 5.97 | 54.59 | 19.75 |
| **PiSSA** | 5.30 | 55.42 | 19.52 |
| **MiLoRA** | 5.23 | 54.44 | 19.51 |
| *LoRA MoE Methods* | | | |
| **MoLoRA** | 5.84 | 56.63 | 24.83 |
| **HydraLoRA** | 5.82 | 57.39 | 24.21 |
| **GOAT** | **6.01** | **60.20** | **25.61** |

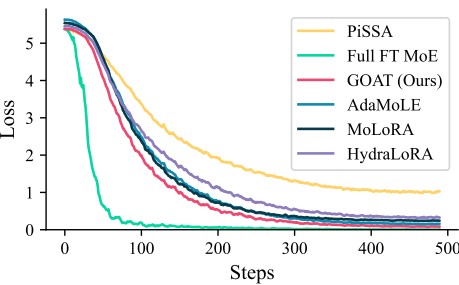

*Figure 4.* Training loss curves of Different LoRA methods and Full Fine-tuning MoE on Cars. The balance loss is excluded in the MoE baselines for a fair comparison with single LoRA baselines.

KASA, by 1.47%, the best LoRA-MoE method, HydraLoRA, by 1.98%, and ChatGPT by 7.42%.

- **NLU** (Table 4): our method outperforms the best-performing Single LoRA Method, MiLoRA, by 0.28%, surpasses the best-performing LoRA MoE Method, MoLoRA, by 0.27%, and achieves a 1.98% improvement over HydraLoRA. Furthermore, our method surpasses the Full FT (89.47 *vs.* 89.76) and reduces the gap with Full FT MoE to just 0.1%.

In summary, GOAT outperforms across all benchmarks, achieving superior results in nearly every sub-task, and closes or surpasses the performance gap with Full FT, demonstrating the superior effectiveness of our approach.

### 4.4. Ablation Study

We conduct ablation experiments to evaluate the impact of our adaptive priors initialization and gradient scaling, as summarized in Table 5. Our initialization, with or without MoE scaling, consistently outperforms other methods[5] (note that no SVD-based initialization corresponds to the

---

[5]Details provided in Appendix G.3

original zero initialization, yielding 81.06/80.26). Without MoE, initializing a single LoRA with our SVD fragments achieves a performance of 77.62. In contrast, our MoE architecture achieves 80.35, demonstrating its clear advantage in effectively integrating expert functionalities.

### 4.5. Convergence Speed

As shown in Figure 4, we compare the training loss curves of PiSSA, various LoRA MoE baselines, our proposed GOAT, and Full FT MoE on the Cars and MetaMathQA datasets. GOAT demonstrates faster convergence compared to the LoRA MoE baselines and achieves performance closest to Full FT MoE. Notably, our method achieves a lower final loss, balancing performance and efficiency. In contrast, methods like PiSSA converge quickly initially but yield suboptimal final performance, as discussed in Section 2.1.

### 4.6. Scaling Property

**Scaling across Different Rank.** To evaluate the scalability of our method, we increase the rank in GOAT from 8 to 128 on CV benchmarks, as shown in Figure 5. As the rank increases, the performance gap between GOAT and

*Table 3.* Performance comparison of LLaMA2 7B with different methods on eight commonsense reasoning datasets. The symbol †
indicates that the results are taken from (Wang et al., 2024a; Zhong et al., 2024; Si et al., 2024).

| Method | # Params(%) | BoolQ | PIQA | SIQA | HellaSwag | WinoGrande | ARC-e | ARC-c | OBQA | Average |
|---|---|---|---|---|---|---|---|---|---|---|
| ChatGPT † | / | 73.10 | 85.40 | 68.50 | 78.50 | 66.10 | 89.80 | 79.90 | 74.80 | 77.01 |
| *Single LoRA Methods* | | | | | | | | | | |
| LoRA† | 0.84 | 69.80 | 79.90 | 79.50 | 83.60 | 82.60 | 79.80 | 64.70 | 81.00 | 77.61 |
| DoRA† | 0.84 | 71.80 | 83.10 | 79.90 | 89.10 | 83.00 | 84.50 | 71.00 | 81.20 | 80.45 |
| PiSSA† | 0.84 | 67.60 | 78.10 | 78.40 | 76.60 | 78.00 | 75.80 | 60.20 | 75.60 | 73.78 |
| MiLoRA† | 0.84 | 67.60 | 83.80 | 80.10 | 88.20 | 82.00 | 82.80 | 68.80 | 80.60 | 79.24 |
| LoRA-Dash† | 0.84 | 71.00 | 75.70 | 79.30 | 91.10 | 78.60 | 84.20 | 69.80 | 78.80 | 78.56 |
| NEAT† | 0.84 | 71.70 | 83.90 | 80.20 | 88.90 | 84.30 | 86.30 | 71.40 | 83.00 | 81.21 |
| KaSA† | 0.84 | 73.60 | **84.40** | 80.20 | **91.50** | 84.50 | 84.70 | 72.10 | 81.20 | 81.53 |
| *LoRA MoE Methods* | | | | | | | | | | |
| MoLoRA | 0.96 | 73.15 | 83.68 | 80.09 | 74.57 | 85.95 | 87.33 | 72.53 | 86.20 | 80.43 |
| HydraLoRA | 0.84 | 72.78 | 84.06 | 79.68 | 80.34 | 86.66 | 87.12 | 72.35 | 86.00 | 81.12 |
| GOAT | 0.96 | **73.60** | 83.95 | **80.50** | 87.12 | **85.00** | **87.79** | **76.88** | **87.00** | **82.73** |

*Table 4.* Performance comparison of RoBERTa-large with different methods on 7 GLUE tasks. Total rank is set to 32.

| Method | # Params (%) | CoLA | SST-2 | MRPC | QQP | MNLI | QNLI | RTE | Average |
|---|---|---|---|---|---|---|---|---|---|
| Full FT | 100 | 84.27 | 95.98 | 85.29 | 91.58 | 89.83 | 94.49 | 84.84 | 89.47 |
| Full FT MoE | 698 | 86.02 | 96.22 | 85.05 | 92.20 | 90.20 | 95.10 | 84.48 | 89.90 |
| *Single LoRA Methods* | | | | | | | | | |
| LoRA | 4.00 | 83.41 | 95.64 | 83.33 | 90.06 | 89.00 | 93.28 | 84.47 | 88.46 |
| DoRA | 4.00 | 85.33 | 95.99 | 84.07 | 91.24 | 89.52 | 93.54 | 84.48 | 89.17 |
| PiSSA | 4.00 | 69.12 | 95.98 | 82.84 | 91.24 | 88.94 | 93.59 | 73.29 | 85.00 |
| MiLoRA | 4.00 | 84.65 | 96.10 | 86.02 | 91.33 | 89.51 | 94.12 | 84.83 | 89.51 |
| rsLoRA | 4.00 | 83.51 | 95.98 | 86.02 | 90.75 | 88.97 | 93.84 | 84.12 | 89.03 |
| *LoRA MoE Methods* | | | | | | | | | |
| MoLoRA | 4.50 | 83.94 | 96.10 | 87.75 | 91.45 | 89.36 | 93.90 | 84.11 | 89.52 |
| AdaMoLE | 4.56 | 83.99 | 95.76 | **86.03** | **91.48** | 89.21 | 93.64 | 83.75 | 89.12 |
| HydraLoRA | 2.75 | 83.89 | 95.52 | 85.04 | 91.02 | 89.34 | 93.87 | 81.22 | 88.56 |
| GOAT | 4.50 | **86.86** | **96.21** | 84.55 | 91.40 | **89.55** | **94.19** | **85.56** | **89.76** |

*Table 5.* Ablation study of GOAT. "MoE" denotes using the MoE
architecture instead of a single LoRA. "MS" refers to using MoE
scaling. "O", "P", "M", and "R" represent initializations from seg-
ments that selected by ours, with the principal singular value, with
the minor singular value, and are randomly selected, respectively.

| MoE | SVD Initialization | | | | Avg. | Avg. (w/o MS) |
|---|---|---|---|---|---|---|
| | O | P | M | R | | |
| ✓ | ✓ | | | | **81.49** | **80.35** |
| ✓ | | ✓ | | | 81.11 | 80.02 |
| ✓ | | | ✓ | | 81.14 | 80.03 |
| ✓ | | | | ✓ | 81.22 | 80.07 |
| ✓ | | | | | 81.06 | 80.26 |
| | ✓ | | | | / | 77.62 |

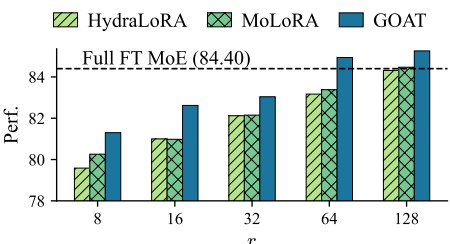

*Figure 5.* Performance of different methods across ranks.

full fine-tuning MoE narrows significantly. Notably, GOAT
consistently outperforms both MoLoRA and HydraLoRA
across all ranks. At rank 32, GOAT achieves 83.04, surpass-
ing MoLoRA (82.15) by 1.08% and HydraLoRA (82.12)
by 1.12%. While higher ranks improve performance, gains
diminish as ranks increase. For instance, GOAT improves
by just 0.38% from rank 64 to 128, highlighting diminishing
returns with higher computational costs.

**Scaling across Different Expert Number and Activated
ratios.** We also conduct experiments on CV datasets fix-

ing total rank as 32 to verify the scalability of our method
with different expert numbers and activation ratios, as shown
in Figure 6. Key findings include: (1) With 8 experts, the
2in8 configuration achieves strong performance. Activat-
ing more experts may yields lower performance, showing
that sparse expert activation is important. (2) Increasing
the total number of experts may improves performance, as
seen in 2in8 vs. 4in16 / 8in32 but makes routers harder to
train, increases memory consumption, and reduces runtime
efficiency. (3) GOAT consistently outperforms MoLoRA,
especially when activate only one expert, consistent with
discussion in Section 2.2. In practice, 2in8 offers a bal-
anced trade-off between performance and storage efficiency.

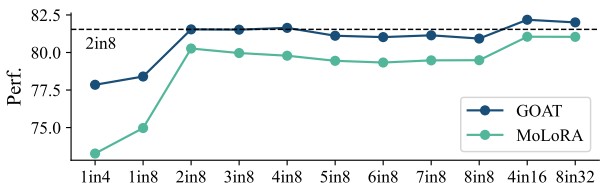

*Figure 6.* Performance vs. number of experts and activation ratio (total rank=32). "2 in 8" means activating 2 out of 8 experts.

### 4.7. Routing Analysis

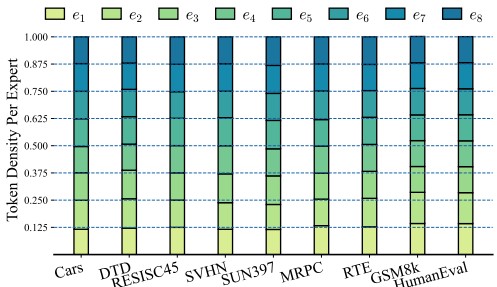

*Figure 7.* Expert Load Distribution across different tasks. We illustrate the fraction of tokens assigned to each expert $\{e_i\}_{i=1}^8$

We visualize the expert load distribution of models trained on 9 tasks in Figure 7. With 8 experts (2 activated), the expected token density is 0.125. The visualization highlights several key observations: (1) The load is evenly distributed, with no inactive experts and fluctuations remaining within 0.125, varying by no more than 15% (0.02). (2) CV and GLUE tasks show balanced expert usage, while generation tasks (GSM8k and HumanEval) favor the bottom-2 experts ($e_1$ and $e_2$) with a load around 0.14. (3) This validates the effectiveness of each SVD chunk, as experts are initialized with distinct singular value regions.

### 4.8. Different Learning Rate

*Table 6.* Performance comparison of different learning rates.

| Learning rate | MoLoRA | HydraLoRA | GOAT |
|---|---|---|---|
| $1e^{-5}$ | 56.18 | 55.19 | **58.74** |
| $2e^{-5}$ | 56.63 | 57.39 | **60.20** |
| $5e^{-5}$ | 60.19 | 60.96 | **62.05** |

To evaluate GOAT's sensitivity to learning rates, we tested its performance on GSM8K using rates ranging from $1 \times 10^{-5}$ to $5 \times 10^{-5}$, comparing it against MoLoRA and HydraLoRA. As shown in Table 6, GOAT consistently outperforms the other methods, showcasing its robustness and the effectiveness of our initialization and scaling strategies in accelerating convergence and enhancing performance.

### 4.9. Computation Analysis

**Parameter Size.** The "# Params (%)" column in Tables 1, 2, 3, and 4 compares the parameter ratios of LoRA baselines and GOAT to full fine-tuning MoE. GOAT achieves state-of-the-art performance with a parameter size of $O(Hr) +$

*Table 7.* Comparison of LoRA-MoE and Full FT MoE in memory cost, training time, and GSM8K performance. Memory cost was measured and training time was recorded on the MetaMath dataset using one A100 GPU with identical batch sizes.

| Method | Memory Cost | Epoch Time | Performance |
|---|---|---|---|
| **Full FT MoE** | $\geq 640$ GB | $\approx 106$h 03min | $\geq 59.36$ |
| MoLoRA | 34.85 GB | 36h56min | 56.63 |
| HydraLoRA | 34.81 GB | 36h56min | 57.39 |
| GOAT | 34.85 GB | 36h59min | 60.20 |

$O(He)$, significantly smaller than Full FT's $O(H^2)$ and Full FT MoE's $O(kH^2)$. Since $r, e \ll H$, GOAT is much more efficient. Detailed analysis is in Appendix H.1.

**FLOPs Analysis** To compare with Full FT MoE, we estimate the memory usage, runtime, and performance of FT MoE based on the single GPU runtime of Full FT. As shown in Table 7, the LoRA-MoE series trains much faster than Full FT MoE. Among LoRA-MoE variants, our method achieves the best performance with identical memory and time costs. FLOPs analysis (see Appendix H.2) reveals that Full FT MoE scales as $O(ksH^2)$, while LoRA MoE simplifies to $O(sH^2)$ since $k < e$ and $r \ll H$. Thus, LoRA MoE's FLOPs remain nearly constant, independent of $k$, unlike Full FT MoE, which scales linearly with $k$.

## 5. Related Work

Since the introduction of LoRA (Hu et al., 2021), various variants have emerged, focusing on three key areas: (1) *Architecture Improvements*: DoRA (Liu et al., 2024) decomposes updates into magnitude and direction, while NEAT (Zhong et al., 2024) introduces nonlinear adaptations. (2) *Adaptive Rank/Scale*, AdaLoRA (Zhang et al., 2023) offers dynamic rank allocation, rsLoRA (Kalajdzievski, 2023) adjusts scaling factors and LoRA+ (Hayou et al., 2024) improves learning rate. (3) *Initialization/Optimization*, PiSSA (Meng et al., 2024), MiLoRA (Wang et al., 2024b), and KaSA (Wang et al., 2024a) utilize SVD-based strategies to preserve knowledges. LoRA-Dash (Si et al., 2024) automates optimal direction discovery, whereas LoRA-GA (Wang et al., 2024c) and LoRA-Pro (Wang et al., 2024d) align updates with full fine-tuning gradients. However, they still exhibit performance gap between full fine-tuning.

Multi-LoRA architectures further boost performance: LoRAHub (Huang et al., 2024) combines task-specific LoRA modules, MoLoRA (Zadouri et al., 2024),MoELoRA (Liu et al., 2023) and LoRAMoE (Dou et al., 2023) integrate MoE structures with LoRA. MultiLoRA (Wang et al., 2023) introduces learnable scaling for each expert, while AdaMoLE (Liu & Luo, 2024) introduces learnable thresholds for dynamic experts selection. HydraLoRA (Tian et al., 2024) adopts an asymmetric MoE architecture. Unlike these methods, GOAT introduces a novel SVD-structured MoE framework that adaptively integrates relevant priors while

addressing weight misalignment and gradient dynamics through theoretical scaling.

## 6. Conclusion

In this work, we propose GOAT, a novel framework that enhances LoRA fine-tuning by adaptively integrating SVD-structured priors and aligning low-rank gradients with full fine-tuned MoE through theoretical scaling. Without altering the architecture or training algorithms, GOAT significantly improves efficiency and performance, achieving state-of-the-art results across 25 diverse datasets. Our approach effectively bridges the performance gap between LoRA-based methods and Full Fine-Tuning.

## Acknowledgments

This work was supported in part by the National Natural Science Foundation of China under Grant No. 62276110, No. 62172039 and in part by the fund of Joint Laboratory of HUST and Pingan Property & Casualty Research (HPL). We thank the Shanghai AI Laboratory for supporting GPU resources. The authors would also like to thank the anonymous reviewers for their comments on improving the quality of this paper.

## Impact Statement

GOAT enhances the efficiency and performance of fine-tuning large models, significantly reducing computational and memory costs. This makes advanced AI technologies more accessible to researchers and practitioners with limited resources, fostering innovation across diverse fields such as NLP, CV, and multi-modal applications. By leveraging adaptive priors and robust gradient handling, GOAT can drive breakthroughs in solving real-world challenges, enabling more efficient and scalable AI solutions for a wide range of industries. Our work focuses on improving model efficiency and adaptability and does not introduce any direct ethical concerns or risks.

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

# A. Pseudocode

**Algorithm 1** GOAT

**Require:** Input vector $x$, input dimension $n$, hyperparameters $\eta, \rho$, number of experts $E$
**Ensure:** Output $y = \tilde{W}_0(x) + \sum_{i=1}^{E} w^i(x)\, s\, B_0^i A_0^i(x)$
 1: **function** Initialization
 2:     **Scaling factor:** $s \leftarrow \sqrt{3n\eta/\rho}$
 3:     **SVD decomposition:** $W_0 = U\Sigma V^\top$
 4:     **for** $i = 1$ to $E$ **do**
 5:        $B_0^i \leftarrow \sqrt{1/(s\rho)}\, U'\Sigma'^{1/2}$
 6:        $A_0^i \leftarrow \sqrt{1/(s\rho)}\, \Sigma'^{1/2} V'^\top$
 7:     **end for**
 8:     $W_{\text{res}}^{+} \leftarrow \frac{s}{E}\sum_{i=1}^{E} B_0^i A_0^i$
 9:     $\tilde{W}_0 \leftarrow W_0 - W_{\text{res}}^{+}$
10:     return: $\tilde{W}_0, \{B_0^i, A_0^i\}$
11: **end function**

 1: **function** Forward($x$)
 2:     Compute gating weight $w^i(x)$ $(1 \le i \le E)$
 3:     return: $\tilde{W}_0(x) + \sum_{i=1}^{E} w^i(x)\, s\, B_0^i A_0^i(x)$
 4: **end function**

# B. Proof related with PiSSA Select Segment

**Lemma B.1.** *Let $W_0 \in \mathbb{R}^{m \times n}$ be the pretrained weight matrix with SVD $W_0 = U\Sigma V^\top$. Assuming $m \le n$ and LoRA rank $r$, we decompose $W_0$ into rank-$r$ blocks:*

$$W_0 = \sum_{i=0}^{l} U_i \Sigma_i V_i^\top, \tag{21}$$

*where $l = \frac{m}{r} - 1$ are block numbers, $U_i = U_{[i\cdot r:(i+1)\cdot r,:]} \in \mathbb{R}^{r \times m}$, $\Sigma_i = \Sigma_{[i\cdot r:(i+1)\cdot r,\, i\cdot r:(i+1)\cdot r]} \in \mathbb{R}^{r \times r}$, and $V_i = V_{[i\cdot r:(i+1)\cdot r,:]} \in \mathbb{R}^{r \times n}$ are submatrices of $U, \Sigma, V$.*
*We demonstrate that $U_0 \Sigma_0 V_0^\top$ has the largest norm and is the best rank-$r$ approximation of $W_0$.*

*Proof.* By the singular value decomposition (SVD), $W_0 = \sum_{i=1}^{\min(m,n)} \sigma_i u_i v_i^\top$, where $\sigma_i$ are singular values sorted in descending order ($\sigma_1 \ge \sigma_2 \ge \cdots$).

For each block $U_i \Sigma_i V_i^\top$, the Frobenius norm can be written as:

$$\|U_i \Sigma_i V_i^\top\|_F = \left\| \sum_{j=i\cdot r}^{(i+1)\cdot r} \sigma_j u_j v_j^\top \right\|_F. \tag{22}$$

Since the Frobenius norm satisfies the property of orthogonal invariance, we can simplify this expression:

$$\|U_i \Sigma_i V_i^\top\|_F = \sqrt{ \sum_{j=i\cdot r}^{(i+1)\cdot r} \sigma_j^2 }. \tag{23}$$

This result shows that the norm of each block $U_i \Sigma_i V_i^\top$ depends solely on the singular values $\sigma_j$ within the block. As the singular values are sorted in descending order ($\sigma_1 \ge \sigma_2 \ge \cdots$), the block $U_0 \Sigma_0 V_0^\top$, which contains the largest $r$ singular values ($\sigma_1, \ldots, \sigma_r$), has the largest Frobenius norm:

$$\|U_0 \Sigma_0 V_0^\top\|_F = \sqrt{ \sum_{j=1}^{r} \sigma_j^2 }. \tag{24}$$

By the Eckart–Young–Mirsky theorem, the best rank-$r$ approximation of $W_0$ minimizes the reconstruction error:

$$\|W_0 - W_0^{(r)}\|_F = \min_{X:\text{rank}(X)\leq r} \|W_0 - X\|_F, \tag{25}$$

where $W_0^{(r)} = U_0\Sigma_0 V_0^\top$. Therefore, $U_0\Sigma_0 V_0^\top$ not only has the largest norm but also preserves the most significant information in $W_0$, making it the optimal rank-$r$ approximation. $\qquad\square$

## C. Load Balance Loss

In vanilla MoE methods (Fedus et al., 2022; Dai et al., 2024), a balance loss $\mathcal{L}_b$ mitigates routing collapse by ensuring even token distribution among experts:

$$\mathcal{L}_b = \sum_{i=1}^{E} f_i P_i \tag{26}$$

$$f_i = \frac{E}{kT} \sum_{t=1}^{T} \mathbb{1}(\text{Token } x_t \text{ assigned to expert } i) \tag{27}$$

$$P_i = \frac{1}{T} \sum_{t=1}^{T} \text{softmax}(z^i(x_t)) \tag{28}$$

where $T$ is the number of tokens and $\mathbb{1}(\cdot)$ is the indicator function. Here, $f_i$ is the fraction of tokens assigned to expert $i$, and $P_i$ is the average routing probability for expert $i$. This loss promotes an even distribution of tokens across experts.

## D. Proof of Theoretical Results

### D.1. Proof of Lemma 2.2

> *Lemma* (2.2). Let $g_t$ be the full-tuning gradient, and $B$, $A$ be low-rank weights. At the $t$-th optimization step, the equivalent gradient can be expressed as:
>
> $$\tilde{g}_t = s^2 \left( B_t B_t^\top g_t + g_t A_t^\top A_t \right) \tag{29}$$

*Proof.* According to the assumption, $\tilde{W}_t = W_t$. Let LoRA $sBA$ where $B \in \mathbb{R}^{m \times r}, A \in \mathbb{R}^{r \times n}$, $s \in \mathbb{R}$, the loss $\mathcal{L}$, the $t^{th}$ update of SGD optimizer. We denote $\tilde{W}_t = W_{\text{init}} + sB_t A_t$, we can write the gradient of $B$, $A$ as:

$$G_t^B = \frac{\partial L}{\partial \tilde{W}_t} \frac{\partial \tilde{W}_t}{\partial B} = \frac{\partial L}{\partial W_t} \frac{\partial \tilde{W}_t}{\partial B} = sg_t A^\top \tag{30}$$

$$G_t^A = \frac{\partial L}{\partial \tilde{W}_t} \frac{\partial \tilde{W}_t}{\partial A} = \frac{\partial L}{\partial W_t} \frac{\partial \tilde{W}_t}{\partial A} = sB^\top g_t \tag{31}$$

In the gradient descend algorithm (SVD), the updates for $B_t$ and $A_t$ are

$$dB_t = -\eta G_t^B = -s\eta g_t A_t^\top, dA_t = -\eta G_t^A = -s\eta B_t^\top g_t \tag{32}$$

The change in the equivalent weight $\tilde{W}$ can be expressed as:

$$d\tilde{W} = \frac{\partial \tilde{W}_t}{\partial A_t} dA_t + \frac{\partial \tilde{W}_t}{\partial B_t} dB_t \tag{33}$$

$$= s \cdot B_t dA_t + s \cdot dB_t A_t \tag{34}$$

$$= s \left( B_t(-\eta s B_t^\top g_t) + (-\eta s g_t A_t^\top) A_t \right) \tag{35}$$

$$= -\eta s^2 \left( B_t B_t^\top g_t + g_t A_t^\top A_t \right) \tag{36}$$

Therefore, the equivalent gradient $\tilde{g}_t$ is given by:

$$\tilde{g}_t = s^2 \left( B_t B_t^\top g_t + g_t A_t^\top A_t \right) \tag{37}$$

This concludes the proof. $\qquad\square$

## D.2. Proof of Theorem 3.1

> *Theorem* (3.1). Let the learning rate in Full FT and LoRA be $\eta_{\text{FFT}}, \eta_{\text{LoRA}}$. By ensuring equivalent weight $\tilde{W}_0 \approx W_0$ at initialization and maintaining equivalent gradient $\eta_{\text{LoRA}}\tilde{g}_t \approx \eta_{\text{FFT}}g_t$ throughout each optimization step, we can effectively align LoRA with Full FT. (Equivalent weight and gradient are defined in Definition 2.1.)

*Proof.* We verify this alignment using induction. The equivalent weight is defined as $\tilde{W}_t = W_{\text{init}} + sB_tA_t$, and the equivalent gradient is $\tilde{g}_t = \frac{\partial L}{\partial \tilde{W}}$. Using the gradient descent algorithm (considering only the SGD optimizer), we have:

$$W_{t+1} = W_t - \eta_{\text{FFT}}g_t \tag{38}$$

$$\tilde{W}_{t+1} = \tilde{W}_t - \eta_{\text{LoRA}}\tilde{g}_t \tag{39}$$

*Base Case ($t = 0$):* We have ensured $\tilde{W}_0 = W_0$.

*Inductive Step:* Assume $\tilde{W}_t = W_t$ and $\tilde{g}_t = g_t$. Then:

$$\tilde{W}_{t+1} = \tilde{W}_t - \eta_{\text{LoRA}}\tilde{g}_t \tag{40}$$

$$= W_t - \eta_{\text{FFT}}g_t \tag{41}$$

$$= W_{t+1}. \tag{42}$$

By induction, $\tilde{W}_t = W_t$ for all $t$, ensuring the alignment between LoRA and Full FT. $\qquad\square$

## D.3. Proof of Theorem 3.2

> *Theorem* (3.2). Let the learning rate in Full FT MoE and LoRA MoE be $\eta_{\text{FFT}}, \eta_{\text{LoRA}}$. For all $i \in [1, \dots, E]$, by ensuring the equivalent weight of the $i$-th expert $\tilde{W}_0^i \approx W_0^i$ at initialization and maintaining the equivalent gradient of the $i$-th expert $\eta_{\text{LoRA}}\tilde{g}_t^i \approx \eta_{\text{FFT}}g_t^i$ throughout each optimization step, we can effectively align LoRA MoE with Full FT MoE.

*Proof.* We aim to show that under the given conditions, the LoRA MoE aligns with the Full FT MoE by effectively making the MoE routers behave identically in both models.

*Base Case ($t = 0$):* At initialization, by assumption, the equivalent weights of each expert satisfy $\tilde{W}_0^i \approx W_0^i$ because our Full FT MoE is an upcycling MoE which makes all $W_0^i = W_0$. Additionally, since both models use the same random seed, the routers are initialized identically, ensuring that the routing decisions are the same for both Full FT MoE and LoRA MoE.

*Inductive Step:* Assume that at step $t$, the equivalent weights satisfy $\tilde{W}_t^i = W_t^i$ for all $i$, and the routers in both models are identical. During the $t$-th optimization step, the gradients are scaled such that $\eta_{\text{LoRA}}\tilde{g}_t^i \approx \eta_{\text{FFT}}g_t^i$. This ensures that the weight updates for each expert in both models are equivalent:

$$\tilde{W}_{t+1}^i = \tilde{W}_t^i - \eta_{\text{LoRA}}\tilde{g}_t^i \approx W_t^i - \eta_{\text{FFT}}g_t^i = W_{t+1}^i \tag{43}$$

First, as the routers are identical, the router weight $w^i$ is the same, so the layer output is the same:

$$\text{MoE}(\mathbf{x}) = \sum_{i=1}^{E} w^i(\mathbf{x}) W^i(\mathbf{x}) \tag{44}$$

$$= \sum_{i=1}^{E} w^i(\mathbf{x}) \tilde{W}^i(\mathbf{x}) \tag{45}$$

$$= \sum_{i=1}^{E} w^i(\mathbf{x}) (W + s B^i A^i)(\mathbf{x}) \tag{46}$$

$$= W(\mathbf{x}) + \sum_{i=1}^{E} w^i(\mathbf{x}) \left( s B^i A^i(\mathbf{x}) \right) \tag{47}$$

$$= \text{MoE}_{\text{LoRA}}(\mathbf{x}) \tag{48}$$

Since the weight updates are equivalent and the routers are optimized from the output induced by these weights, the routers remain identical at step $t + 1$. Therefore, by induction, the routers are identical for all $t$.

With identical routers, the routing decisions do not differentiate between Full FT MoE and LoRA MoE layers. Consequently, the alignment of individual experts (as established by Theorem 3.1) ensures that the overall behavior of both MoE variants is effectively aligned.

□

## D.4. Proof of Lemma 3.3

*Lemma* (3.3). Let $\Omega_k(\mathbf{x})$ be the set of indices corresponding to the top-$k$ largest values of $z^i(\mathbf{x})$, and $z^i(\mathbf{x})$ are independent and identically distributed (i.i.d.), and $k \le \frac{E}{2}$, $w^i$ is defined as:

$$w^i(\mathbf{x}) = \begin{cases} \frac{\exp(z^i(\mathbf{x}))}{\sum_{j \in \Omega_k(\mathbf{x})} \exp(z^j(\mathbf{x}))} & \text{if } i \in \Omega_k(\mathbf{x}), \\ 0 & \text{if } i \notin \Omega_k(\mathbf{x}), \end{cases} \tag{49}$$

We demonstrate the following properties for all $i, j \in [1, \ldots, E]$ $(i \neq j)$:

$$\mathbb{E}_{\mathbf{x}}[w^i(\mathbf{x})] = \frac{1}{E}, \tag{50}$$

$$\text{Var}_{\mathbf{x}}(w^i(\mathbf{x})) = \frac{E - k}{k E^2}. \tag{51}$$

*Proof.* Because the $z^i(x)$ are i.i.d. random variables, any permutation of the indices $\{1, \ldots, E\}$ leaves the joint distribution of $\{z^1(\mathbf{x}), \ldots, z^E(\mathbf{x})\}$ unchanged. The Top-K operation (pick the indices of the largest $K$ logits) is also symmetric with respect to permutations: permuting $(z^1, \ldots, z^E)$ accordingly permutes the set $\Omega_k(\mathbf{x})$ of selected indices. Because of this symmetry, each $w^i(\mathbf{x})$ is distributed in the same way as $w^j(\mathbf{x})$ for any $j$. By definition of $w^i(\mathbf{x})$, we have $\forall \mathbf{x}, \sum_{i=1}^{E} w^i(\mathbf{x}) = 1$, so:

$$\sum_{i=1}^{E} \mathbb{E}[w^i(\mathbf{x})] = \mathbb{E}\left[ \sum_{i=1}^{E} w^i(\mathbf{x}) \right] = \mathbb{E}[1] = 1, \tag{52}$$

$$\mathbb{E}_{\mathbf{x}}[w^i(\mathbf{x})] = \frac{1}{E}, \forall i \in [1, \cdots, E] \tag{53}$$

The variance of $w^i(\mathbf{x})$ is given by:

$$\text{Var}_{\mathbf{x}}(w^i(\mathbf{x})) = \mathbb{E}_{\mathbf{x}}\left[ \left( w^i(\mathbf{x}) \right)^2 \right] - \left( \mathbb{E}_{\mathbf{x}}\left[ w^i(\mathbf{x}) \right] \right)^2. \tag{54}$$

Since $\mathbb{E}_{\mathbf{x}}\left[w^i(\mathbf{x})\right] = \frac{1}{E}$, we have:

$$\text{Var}_{\mathbf{x}}(w^i(\mathbf{x})) = \mathbb{E}_{\mathbf{x}}\left[\left(w^i(\mathbf{x})\right)^2\right] - \frac{1}{E^2}. \tag{55}$$

We aim to compute $\mathbb{E}_{\mathbf{x}}\left[\left(w^i(\mathbf{x})\right)^2\right]$, but it's tricky to directly obtain this expectation. Given that $\sum_{i=1}^{E} w_i = 1$, we can expand this expression. Omitting the $\mathbf{x}$ for simplicity, we get:

$$1 = \left(\sum_{i=1}^{E} w_i\right)^2 = \mathbb{E}\left[\left(\sum_{i=1}^{E} w_i\right)^2\right] = \mathbb{E}\left[\sum_{i=1}^{E} w_i^2\right] + \sum_{i \neq j} \mathbb{E}[w_i w_j], \tag{56}$$

$$1 = E \cdot \mathbb{E}[w_i^2] + E(E-1) \cdot \mathbb{E}_{i \neq j}[w_i w_j]. \tag{57}$$

where $\mathbb{E}[w_i w_j]$ is the expectation we need to compute. This expression is derived based on the rotational symmetry of $w_i, w_j$, which means the cross-term $\mathbb{E}[w_i w_j]$ is the same for all distinct $i \neq j$.

To compute $\mathbb{E}[w_i w_j]$, we rewrite the weights $w_i$ as follows:

$$w_i = \frac{\exp z_i}{\sum_{j \in \Omega_k} \exp z_j} = \frac{y_i}{\sum_{j \in \Omega_k} y_j}, \tag{58}$$

where

$$y_i = \begin{cases} \exp z_i & \text{if } i \in \Omega_k(\mathbf{x}), \\ 0 & \text{if } i \notin \Omega_k(\mathbf{x}). \end{cases} \tag{59}$$

Thus, the product $w_i w_j$ becomes:

$$w_i w_j = \frac{y_i y_j}{\left(\sum_{j \in \Omega_k} y_j\right)^2}. \tag{60}$$

Now, due to rotational symmetry of the terms $y_i, w_j$, we can compute:

$$\mathbb{E}[w_i w_j] = \frac{\binom{k}{2}}{\binom{E}{2}} \mathbb{E}\left[\frac{y_i y_j}{\left(\sum_{j \in \Omega_k} y_j\right)^2}\right] = \frac{k(k-1)}{E(E-1)} \cdot \frac{1}{k^2} = \frac{k-1}{E(E-1)k}. \tag{61}$$

Substituting this back into Equation (57) for $\mathbb{E}[w_i^2]$:

$$1 = E \cdot \mathbb{E}[w_i^2] + E(E-1) \cdot \frac{k-1}{E(E-1)k}, \tag{62}$$

we get:

$$\mathbb{E}[w_i^2] = \frac{1}{Ek}. \tag{63}$$

Thus, the variance of $w^i$ in Equation (55) is:

$$\text{Var}(w^i) = \frac{1}{Ek} - \frac{1}{E^2} = \frac{E-k}{kE^2}. \tag{64}$$

$\square$

## D.5. Proof of Theorem 3.4

*Theorem* (3.4). Consider the optimization problem:

$$W_{\text{res}}^+ = \arg\min_{W_{\text{res}}} \mathbb{E}_{\mathbf{x}} \left[ \left\| W_{\text{res}} - s \sum_{i=1}^E w^i(\mathbf{x}) B_0^i A_0^i \right\|^2 \right]. \tag{65}$$

The closed-form solution is $W_{\text{res}}^+ = \frac{s}{E} \sum_{i=1}^E B_0^i A_0^i$.

*Proof.* $W_{res}^+$ denotes the optimal value of $W_{res}$. The solution to this optimization problem, $W_{res}$, can be derived as the expected value over all possible $\mathbf{x}$:

$$W_{\text{res}}^+ = s\mathbb{E}_{\mathbf{x}} \left[ \sum_{i=1}^E w^i(\mathbf{x}) B_0^i A_0^i \right] \tag{66}$$

$$= s \sum_{i=1}^E \mathbb{E}_{\mathbf{x}}[w^i(\mathbf{x})] B_0^i A_0^i \tag{67}$$

$$= \frac{s}{E} \sum_{i=1}^E B_0^i A_0^i \tag{68}$$

where Equation (66) use the linear property of expectation and Equation (67) utilize Lemma 3.3.

$\square$

## D.6. Proof of Theorem 3.5

*Theorem* (3.5). Consider the optimization problem where $B_0 = 0$ and $A_0 \sim U\left(-\sqrt{\frac{6}{n}}, \sqrt{\frac{6}{n}}\right)$, $\tilde{g}_t^i = s^2 \left( B_t^i B_t^{i\top} g_t^i + g_t^i A_t^{i\top} A_t^i \right)$, the ratio between full tuning learning rate *vs.* LoRA learning rate $\eta$.

$$\arg\min_s \left\| \tilde{g}_t^i - g_t^i \right\|, \quad \forall i \in [1, \dots, E] \tag{69}$$

The closed-form solution is $s = \sqrt{\frac{3n\eta}{r}}$.

*Proof.* By analyzing the first step gradient,

$$\tilde{g}_0 = s(B_0 G_0^A + G_0^B A_0) = s^2(B_0 B_0^\top g_0 + g_0 A_0^\top A_0) \tag{70}$$

$$\arg\min_s \left\| s^2 \underbrace{\left( B_0 B_0^\top g_0 + g_0 A_0^\top A_0 \right)}_{\text{rank}<2r} - \eta g_0 \right\| \tag{71}$$

As LoRA init $B_0 = 0$ and $A_0 \sim U(-\sqrt{\frac{6}{n}}, \sqrt{\frac{6}{n}})$. The above equation becomes

$$\arg\min_s \left\| s^2 \underbrace{\left( g_0 A_0^\top A_0 \right)}_{\text{rank}<2r} - \eta g_0 \right\| \tag{72}$$

First We notice that the matrix $A_0^\top A_0$ can express the entries in the following way

$$A_0^\top A_0[i,j] = \sum_{k=1}^{r} A_0[i,k]A_0^\top[k,j], \tag{73}$$

For the diagonal entries $(i=j)$, the formula simplifies to:

$$(A_0^\top A_0)_{i,i} = \sum_{k=1}^{r} A_{0,i,k}^2 = \sigma_A \tag{74}$$

This is because the entries of $A_0$ are i.i.d. with mean 0 and variance $\sigma_A$, we can compute:

$$\mathbb{E}[(A_0^\top A_0)_{i,i}] = \sum_{k=1}^{r} \mathbb{E}[A_{0,i,k}^2] = r\sigma_A \tag{75}$$

For the non-diagonal entries $(i \neq j)$, the formula is:

$$(A_0^\top A_0)_{i,j} = \sum_{k=1}^{r} A_0^\top[i,k]A_0[k,j] = 0 \tag{76}$$

Since $A_0^\top[i,k]$ and $A_0[k,j]$ are independent random variables (for $i \neq j$), their product has an expected value of zero.

$$\mathbb{E}_{A_0}[A_0^\top A_0] = r\sigma_A \mathbf{I}_{n \times n} \tag{77}$$

Given that $\mathbb{E}_{A_0}[A_0^\top A_0] = \frac{r}{3n}\mathbf{I}_{n \times n}$ (use Leaky ReLU (Xu et al., 2015) with negative slope $\sqrt{5}$, that is $\mathrm{Var}(A) = \frac{1}{3n}$), we can get $s = \sqrt{\frac{3n\eta}{r}}$

$$\left\| g_0 \left( \frac{s^2 r}{3n}\mathbf{I} - \eta\mathbf{I} \right) \right\| = 0, \quad s = \sqrt{\frac{3n\eta}{r}} \tag{78}$$

Though it is derived by the first step gradient, as in practice, the weight change $\|\frac{\mathrm{d}W}{W}\|$ is typically small (thus has the low-rank update hypnosis in Hu et al. (2021)), we can consider $\|\frac{\mathrm{d}A}{A}\|$ and $\|\frac{\mathrm{d}B}{B}\|$ is small, so the above $s$ can be extended to the following steps.

$\square$

## E. Extend Our Method to Scenarios with Proper Scaling

GOAT assumes a scenario where LoRA MoE has not been properly scaled. Here, we supplement it with an extended approach for scenarios where proper scaling has been applied.

Here, we assume that the routing strategy of the fully fine-tuned MoE aligns with our method. Since the router is non-differentiable, we ignore its impact and focus solely on the gradient of each expert. Our goal is to align the gradient of each expert in our method with that of the fully fine-tuned MoE. Thus, for the $i$-th expert, we aim to solve:

$$\arg\min_{s_i} \left\| s_i^2 \underbrace{\left( B_0^i B_0^{i\top} g_0^i + g_0^i A_0^{i\top} A_0^i \right)}_{\text{rank}<2r} - g_0^i \right\| \tag{79}$$

When using the balanced initialization strategy, the above equation can be rewritten as:

$$\arg\min_{s_i} \left\| s_i^2 \underbrace{\left( u_i u_i^\top \sigma_i^2 g_0^i + g_0^i \sigma_i^2 v_i^\top v_i \right)}_{\text{rank}<2r} - g_0^i \right\| \tag{80}$$

If each expert has rank 1, the equation can be further simplified to:

$$\arg\min_{s_i} \left\| s_i^2 \sigma_i^2 \underbrace{\left( u_i u_i^\top g_0^i + g_0^i v_i^\top v_i \right)}_{\text{rank} < 2r} - g_0^i \right\| \tag{81}$$

From this, we can observe that $\sigma_i$ **acts as a scaling factor for the gradient, stretching or compressing the direction represented by the current expert during optimization.** Here, we assume that the hyperparameters have already been correctly scaled for the first expert (which corresponds to the optimal low-rank approximation of the original matrix), aligning it with the first expert of the fully fine-tuned MoE. Since the stretching strategy for the direction represented by each expert should remain consistent during MoE fine-tuning, we need to align the scaling factors $s_i$ for the other experts to reduce the gap between our method and full MoE fine-tuning. Specifically, $s_i$ must satisfy the following condition:

$$s_1^2 \sigma_0 = s_i^2 \sigma_i \tag{82}$$

Thus, we transform each $s_i$ as follows:

$$s_i = s_1 \frac{\sqrt{\sigma_0}}{\sqrt{\sigma_i}} \tag{83}$$

When the rank of each expert is greater than 1, we approximate the solution by using the sum of the singular values within the segment.

Here, we modify the scaling of all experts except the first one, while keeping other initialization methods consistent with 19.

We refer to this extended method as GOAT-s, and its performance across all benchmarks is presented in Table 8. While designed for different scenarios, it demonstrates performance comparable to GOAT.

*Table 8.* Performance comparison of our method extended to properly scaled scenarios.

| Method | NLG(Avg.) | NLU(Avg.) | IC(Avg.) | CR(Avg.) | Avg. |
|--------|-----------|-----------|----------|----------|------|
| **GOAT** | 30.60 | 89.76 | 81.49 | 82.64 | 71.12 |
| **GOAT-s** | 30.54 | 89.61 | 81.54 | 82.41 | 71.02 |

# F. Further Discussion

## F.1. Discussion on the Practical Applications and Real-world Impact of LoRA MoE

MoE is popular for managing large parameters while activating only a sparse subset during inference, making it ideal for large-scale models. However, in Section 4.9 and Table 7, without optimization, fully fine-tuning an MoE model significantly increases trainable parameters and FLOPs compared to Full FT.

LoRA MoE addresses these challenges by replacing experts with low-rank matrices, reducing computation, preserving MoE benefits, and enabling faster training, lower memory usage, and reduced energy consumption—crucial for resource-limited or real-time applications.

For instance, in NLP, where large-scale models are common, LoRA MoE achieves SOTA performance with lower computational cost. This efficiency benefits industries like autonomous driving, healthcare (Tian et al., 2024), where lower latency and costs enhance performance and scalability.

Overall, LoRA MoE balances MoE's model capacity with cost-effective deployment, making it adaptable to various real-world applications.

## F.2. Analysis of the Phenomenon Where PiSSA and MiLoRA May Perform Worse Than LoRA in Experiments

Previous works (Wang et al., 2024b;a) show that PiSSA and MiLoRA don't always outperform LoRA. KaSA found that PiSSA accelerates convergence but uses limited pre-trained knowledge at lower ranks, limiting performance. Similarly, MiLoRA's minimal adjustments to pre-trained weights often fail to improve over LoRA. In Table 1, we adopt the same rank settings as KaSA and reach the same conclusion.

In contrast, our method consistently achieves superior performance across both low and high ranks by effectively balancing convergence speed and final performance, as demonstrated in Tables 1,2,3,4 and Figure 5.

## F.3. Compared to Other Routing Techniques

As shown in Table 9, we extend our experiments to include alternative routing strategies such as top-p routing and a top-k variant with shared experts. We find that, compared to other approaches, setting $k = 2$ achieves the best performance. We will incorporate these into our revised version of the paper.

*Table 9.* Comparison of Routing Strategies on Average Accuracy

| Routing Strategy | Avg. ACC (%) |
|---|---|
| Ours (top-$k = 2$) | **81.49** |
| Top-$p$ ($p = 0.25$) | 79.40 |
| Top-$k$ + Shared Expert | 78.68 |

## F.4. Analysis of the coefficient for the balance loss

We use top-k routing with k=2 and set the coefficient for the balance loss to 1e-3. As shown in Table 10, we attach the load-balancing loss coefficient experiment by activating 2 out of 8 experts on Cars. We can observe that setting the coefficient

*Table 10.* Performance Comparison under Different Coefficient Values

| Coefficient | GOAT | MoLoRA | HydraLoRA |
|---|---|---|---|
| $1 \times 10^{-1}$ | 49.09 | 49.02 | 48.45 |
| $1 \times 10^{-2}$ | 50.52 | 49.33 | 49.45 |
| $1 \times 10^{-3}$ | 53.50 | 50.83 | 48.42 |
| $1 \times 10^{-4}$ | 51.53 | 49.03 | 48.52 |
| 0 | 49.85 | 48.02 | 49.06 |

too low (e.g., 0 or 1e-4) leads to expert imbalances, which in turn degrades performance. Conversely, excessively high coefficients (e.g., 0.01 or 0.1) can disrupt the normal learning process. Our results show that a coefficient of 1e-3 achieves the best tradeoff in GOAT/MoLoRA between balancing expert load and maintaining stable learning.

Notably, GOAT consistently outperforms across all tested coefficients, demonstrating its robustness in these settings.

## F.5. Is expert activation balanced without the coefficient for the balance loss?

*Table 11.* Expert Activation Distribution of GOAT without Load Balancing

| Method | f1 | f2 | f3 | f4 | f5 | f6 | f7 | f8 |
|---|---|---|---|---|---|---|---|---|
| GOAT w/o Load Balance | 0.1043 | 0.1379 | 0.1275 | 0.1094 | 0.1207 | 0.1405 | 0.1259 | 0.1338 |

We further ablate the load-balancing component in GOAT (2in8, Cars task) and observe that all experts remain active (see Table 11), indicating that each SVD chunk contributes meaningfully to the final representation.

# G. Experiment Details

## G.1. Dataset details

**Natural Language Understanding Tasks.** We evaluate our model on the following natural language understanding tasks from the GLUE benchmark (Wang et al., 2019):

1. **CoLA** (Warstadt et al., 2019): A binary classification task that requires determining whether a given sentence is grammatically acceptable.
2. **SST-2** (Socher et al., 2013): A sentiment analysis task where the goal is to classify sentences as expressing positive or negative sentiment.
3. **MRPC** (Dolan & Brockett, 2005): A binary classification task focused on identifying whether two sentences in a pair are semantically equivalent.
4. **QQP** (Wang et al., 2017): A binary classification task to determine whether two questions from Quora have the same meaning.
5. **MNLI** (Williams et al., 2018): A textual entailment task that involves predicting whether a hypothesis is entailed, contradicted, or neutral with respect to a given premise.
6. **QNLI** (Rajpurkar et al., 2016): A binary classification task to determine whether a question is answerable based on a given context.
7. **RTE** (Giampiccolo et al., 2007): A textual entailment task where the goal is to predict whether a hypothesis logically follows from a given premise.

We report the overall accuracy (including matched and mismatched) for MNLI, Matthew's correlation coefficient for CoLA, and accuracy for all other tasks.

**Natural Language Generation Tasks.** We evaluate our model on the following natural language generation tasks:

1. **MT-Bench** (Zheng et al., 2023): A benchmark for evaluating dialogue generation capabilities, focusing on multi-turn conversational quality and coherence.
2. **GSM8K** (Cobbe et al., 2021): A mathematical reasoning task designed to assess the model's ability to solve grade school-level math problems.
3. **HumanEval** (Chen et al., 2021): A code generation benchmark that measures the model's ability to write functional code snippets based on natural language problem descriptions.

Following previous work (Wang et al., 2024c), we evaluate three natural language generation tasks—dialogue, mathematics, and code—using the following three datasets for training:

1. **Dialogue: WizardLM** (Xu et al., 2023): WizardLM leverages an AI-driven approach called Evol-Instruct. Starting with a small set of initial instructions, Evol-Instruct uses an LLM to rewrite and evolve these instructions step by step into more complex and diverse ones. This method allows the creation of large-scale instruction data with varying levels of complexity, bypassing the need for human-generated data. We use a 52k subset of WizardLM to train our model for dialogue task (MT-bench).
2. **Math: MetaMathQA** (Yu et al.): MetaMathQA is a created dataset designed specifically to improve the mathematical reasoning capabilities of large language models. We use a 100k subset of MetaMathQA to train our model for math task (GSM8K).
3. **Code: Code-Feedback** (Zheng et al., 2024): This dataset includes examples of dynamic code generation, execution, and refinement guided by human feedback, enabling the model to learn how to improve its outputs iteratively. We use a 100k subset of Code-Feedback to train our model for code task (HumanEval).

We evaluate performance on GSM8K using Exact Match, HumanEval using Pass@1, and MT-Bench using the First-Turn Score assessed by GPT-4.

**Image Classification Tasks.** We evaluate our model on the following image classification tasks:

1. **SUN397** (Xiao et al., 2016): A large-scale scene classification dataset containing 108,754 images across 397 categories, with each category having at least 100 images.
2. **Cars** (Stanford Cars) (Krause et al., 2013): A car classification dataset featuring 16,185 images across 196 classes,

evenly split between training and testing sets.

3. **RESISC45** (Cheng et al., 2017): A remote sensing scene classification dataset with 31,500 images distributed across 45 categories, averaging 700 images per category.
4. **EuroSAT** (Helber et al., 2019): A satellite image classification dataset comprising 27,000 geo-referenced images labeled into 10 distinct classes.
5. **SVHN** (Netzer et al., 2011): A real-world digit classification dataset derived from Google Street View images, including 10 classes with 73,257 training samples, 26,032 test samples, and 531,131 additional easy samples.
6. **GTSRB** (Stallkamp et al., 2011): A traffic sign classification dataset containing over 50,000 images spanning 43 traffic sign categories.
7. **DTD** (Cimpoi et al., 2014): A texture classification dataset with 5,640 images across 47 classes, averaging approximately 120 images per class.

We report the accuracy in all tasks.

**Commonsense Reasoning Tasks**   We evaluate our model on the following commonsense reasoning tasks:

1. **BoolQ** (Clark et al., 2019): A binary question-answering task where the goal is to determine whether the answer to a question about a given passage is "yes" or "no."
2. **PIQA** (Physical Interaction Question Answering) (Bisk et al., 2020): Focuses on reasoning about physical commonsense to select the most plausible solution to a given problem.
3. **SIQA** (Social IQa) (Sap et al., 2019): Tests social commonsense reasoning by asking questions about motivations, reactions, or outcomes in social contexts.
4. **HellaSwag** (Zellers et al., 2019): A task designed to test contextual commonsense reasoning by selecting the most plausible continuation of a given scenario.
5. **WinoGrande** (Sakaguchi et al., 2021): A pronoun coreference resolution task that requires reasoning over ambiguous pronouns in complex sentences.
6. **ARC-e** (AI2 Reasoning Challenge - Easy) (Clark et al., 2018): A multiple-choice question-answering task focused on elementary-level science questions.
7. **ARC-c** (AI2 Reasoning Challenge - Challenge) (Clark et al., 2018): A more difficult subset of ARC, containing questions that require advanced reasoning and knowledge.
8. **OBQA** (OpenBookQA) (Mihaylov et al., 2018): A question-answering task requiring reasoning and knowledge from a small "open book" of science facts.

We report the exact match accuracy in all tasks.

## G.2. Baseline details

### Full-Finetune

1. **Full FT** refers to fine-tuning the model with all parameters.
2. **Full FT MoE** refers to fine-tuning all parameters within a Mixture of Experts (MoE) architecture.

### Single-LoRA baselines

1. **LoRA** (Hu et al., 2021) introduces trainable low-rank matrices for efficient fine-tuning.
2. **DoRA** (Liu et al., 2024) enhances LoRA by decomposing pre-trained weights into magnitude and direction, fine-tuning the directional component to improve learning capacity and stability.
3. **PiSSA** (Meng et al., 2024) initializes LoRA's adapter matrices with the principal components of the pre-trained weights, enabling faster convergence, and better performance.
4. **MiLoRA** (Wang et al., 2024b) fine-tunes LLMs by updating only the minor singular components of weight matrices, preserving the principal components to retain pre-trained knowledge.
5. **rsLoRA** (Kalajdzievski, 2023) introduces a new scaling factor to make the scale of the output invariant to rank
6. **LoRA-Dash** (Si et al., 2024) enhances PEFT by leveraging task-specific directions (TSDs) to optimize fine-tuning efficiency and improve performance on downstream tasks.
7. **NEAT** (Zhong et al., 2024) introduces a nonlinear parameter-efficient adaptation method to address the limitations of

existing PEFT techniques like LoRA.

8. **KaSA** (Wang et al., 2024a) leverages singular value decomposition with knowledge-aware singular values to dynamically activate knowledge that is most relevant to the specific task.

## LoRA MoE baseliness

1. **MoLoRA** (Zadouri et al., 2024) combines the Mixture of Experts (MoE) architecture with lightweight experts, enabling extremely parameter-efficient fine-tuning by updating less than 1% of model parameters.
2. **AdaMoLE** (Liu & Luo, 2024) introducing adaptive mechanisms to optimize the selection of experts.
3. **HydraLoRA** (Tian et al., 2024) introduces an asymmetric LoRA framework that improves parameter efficiency and performance by addressing training inefficiencies.

## G.3. Abaltion details

Here, we provide a detailed explanation of the construction of each initialization method. Suppose $h = min(m, n), t = \frac{h}{E}$

1. **Ours (O)**: $\mathcal{E}_r = \left\{ (U_{[:,k:k+d]}, \Sigma_{[k:k+d,k:k+d]}, V^\top_{[k:k+d,:]}) \mid k = (j-1)t, j = 1, \ldots, E \right\}$
2. **Principal (P)**: $\mathcal{E}_r = \left\{ (U_{[:,k:k+d]}, \Sigma_{[k:k+d,k:k+d]}, V^\top_{[k:k+d,:]}) \mid k = (j-1)d, j = 1, \ldots, E \right\}$
3. **Minor (M)**: $\mathcal{E}_r = \left\{ (U_{[:,k:k+d]}, \Sigma_{[k:k+d,k:k+d]}, V^\top_{[k:k+d,:]}) \mid k = h - jd, j = 1, \ldots, E \right\}$
4. **Random (R)**: $\mathcal{E}_r = (U_{[:,k:k+d]}, \Sigma_{[k:k+d,k:k+d]}, V^\top_{[k:k+d,:]}) \mid k = tj, t = random(0, \frac{h}{d} - 1), j = 0, ..., E - 1\}$

## G.4. Implementation Details

Image classification and natural language understanding experiments are conducted on 8 Nvidia 4090 GPUs with 24GB of RAM each. Commonsense reasoning and natural language generation experiments are conducted on a single Nvidia A100 GPU with 80GB of RAM. For training and evaluating all models, we enabled bf16 precision.

*Table 12.* Hyperparameters of the commonsense reasoning task for GOAT.

| Hyperparameter | Commonsense Reasoning |
| --- | --- |
| Batch Size | 16 |
| Rank | 32 |
| Alpha | 64 |
| Optimizer | AdamW |
| Warmup Steps | 100 |
| Dropout | 0.05 |
| Learning Rate | 1e-4 |
| Epochs | 3 |

*Table 13.* Hyperparameters of the image classification task for GOAT.

| Hyperparameter | Cars | DTD | EuroSAT | GTSRB | RESISC45 | SUN397 | SVHN |
| --- | --- | --- | --- | --- | --- | --- | --- |
| Batch Size | | | | 512 | | | |
| Rank | | | | 8 | | | |
| Alpha | | | | 16 | | | |
| Optimizer | | | | AdamW | | | |
| Warmup Steps | | | | 100 | | | |
| Dropout | | | | 0.05 | | | |
| Learning Rate | | | | 1e-4 | | | |
| Epochs | 35 | 76 | 12 | 11 | 15 | 14 | 4 |

## G.5. Hyperparameters

We fine-tune our model on each task using carefully selected hyperparameters to ensure optimal performance. Specific details for each task, including learning rate, batch size, number of epochs, and other configurations, are provided to ensure

*Table 14.* Hyperparameters of the natural language understanding tasks for GOAT.

| Hyperparameter | CoLA | SST-2 | MRPC | QQP | MNLI | QNLI | RTE |
|---|---|---|---|---|---|---|---|
| **Batch Size** | | | | 256 | | | |
| **Rank** | | | | 8 | | | |
| **Alpha** | | | | 16 | | | |
| **Optimizer** | | | | AdamW | | | |
| **Warmup Steps** | | | | 100 | | | |
| **Dropout** | | | | 0.05 | | | |
| **Learning Rate** | | | | 1e-4 | | | |
| **Epochs** | 10 | 10 | 10 | 10 | 10 | 10 | 50 |

*Table 15.* Hyperparameters of the natural language generation task for GOAT.

| Hyperparameter | Natural Language Generation |
|---|---|
| **Batch Size** | 32 |
| **Rank** | 8 |
| **Alpha** | 16 |
| **Optimizer** | AdamW |
| **Warmup Steps** | 100 |
| **Dropout** | 0.05 |
| **Learning Rate** | 2e-5 |
| **Epochs** | 5 |

reproducibility and consistency across experiments. These details are summarized in Table 12, Table 13, Table 14 and Table 15. We set $\rho$ to 10. The ratio between the full fine-tuning learning rate and the LoRA learning rate $\eta$ is empirically set to 1 for ViT/RoBERTa. In the LLaMA experiments, when using a learning rate at the $1e-4$ level, we set $\eta = 0.1$; for a learning rate at the $1e-5$ level, we set $\eta = 1$. This configuration aligns with common practice, where LoRA tuning typically uses a learning rate around $1e-4$, while FFT-based methods operate at a lower learning rate near $1e-5$. We set the coefficient for the balance loss to 1e-3 in our LoRA-MoE experiments. In our LoRA-MoE setup, we use a top-k routing strategy with $k = 2$, which as shown in Table 9 and Figure 6, outperforms other strategies. The same routing strategy is also adopted by all LoRA-MoE baselines.

# H. Parameter and FLOPs Analysis

## H.1. Parameter Analysis

Here, we provide a parameter analysis for each baseline and our method based on different backbones. We assume $H$ represents the model dimension, $r$ denotes the rank, $e$ indicates the number of experts, $L$ indicates the number of layers, $V$ indicates the vocabulary size, $P$ indicates the patch size in ViT and $C$ indicates the number of channels in ViT. The analysis for RoBERTa-large, ViT-base, and LLAMA2 7B is as follows:

**RoBERTa-large:** $H = 1024, r = 32, e = 2, L = 24, V = 50265$. The activation parameters are `dense` from all attention and MLP layer.

1. **FFT (Full Fine-Tuning)**:
   - **Total Parameters**: $(12H^2 + 13H)L + VH$
   - **Breakdown**:
     - Embedding layer: $VH$
     - Attention mechanism: $4H^2 + 4H$
     - MLP layer: $8H^2 + 5H$
     - LayerNorm (2 layers): $4H$
     - Total per layer: $12H^2 + 13H$
2. **Full FT MoE**:
   - **Total Parameters**: $(12eH^2 + 2H + 9He)L + VH$

- **Proportion**: $698\%$

3. **LoRA/PiSSA/MiLoRA/rsLoRA**:
    - **Total Parameters**: $18HrL$
    - **Proportion**: $4.00\%$
4. **DoRA**:
    - **Total Parameters**: $(18Hr + 6)L$
    - **Proportion**: $4.00\%$
5. **MoLoRA/GOAT**:
    - **Total Parameters**: $(18Hr + 9He)L$
    - **Proportion**: $4.50\%$
    - **Breakdown**:
        - Attention mechanism: $8Hr + 4He$
        - MLP layer: $10Hr + 5He$
        - Total per layer: $18Hr + 9He$
6. **HydraLoRA**:
    - **Total Parameters**: $(9Hr + 9He + 9Hr/e)L$
    - **Proportion**: $2.75\%$
7. **AdaMoLE**:
    - **Total Parameters**: $(18Hr + 9He + 9H)L$
    - **Proportion**: $4.56\%$

**ViT-base:** $H = 768, r = 8, e = 2, L = 12, P = 32, C = 3$. The activation parameters include `q, k, v, o, fc1, fc2`.

1. **FFT**:
    - **Total Parameters**: $(C + 1)P^2H + (12H^2 + 2H)L + 3H + PH + H^2$
    - **Breakdown**:
        - Embedding layer: $PH + H + (C + 1)P^2H$
        - encoder (L layers): $(12H^2 + 2H)L$
        - LayerNorm (1 layers): $2H$
        - Pooler: $H^2$
2. **Full FT MoE**:
    - **Total Parameters**: $(C + 1)PPH + (12eH^2 + 2H + 9He)L + 3H + PH + H^2$
    - **Proportion**: $770\%$
3. **LoRA/PiSSA/MiLoRA**:
    - **Total Parameters**: $18HrL$
    - **Proportion**: $1.49\%$
4. **LoRA (rank=16)**:
    - **Total Parameters**: $18HrL$
    - **Proportion**: $2.99\%$
5. **LoRA (rank=32)**:
    - **Total Parameters**: $18HrL$
    - **Proportion**: $5.98\%$
6. **DoRA**:
    - **Total Parameters**: $(18Hr + 6)L$
    - **Proportion**: $1.49\%$
7. **MoLoRA/GOAT**:
    - **Total Parameters**: $(18Hr + 9He)L$

- **Breakdown**:
  - Attention mechanism: $8Hr + 4He$
  - MLP layer: $10Hr + 5He$
  - Total per layer: $18Hr + 9He$
- **Proportion**: 2.24%

8. **HydraLoRA**:
  - **Total Parameters**: $(9Hr + 9He + 9Hr/e)L$
  - **Proportion**: 1.58%

9. **AdaMoLE**:
  - **Total Parameters**: $(18Hr + 9He + 9H)L$
  - **Proportion**: 2.33%

**LLAMA2-7B:** $H = 4096, r = 32, e = 2, L = 32, V = 32000$. The activation parameters are `q`, `k`, `v`, `up`, `down`.

1. **FFT**:
  - **Total Parameters**: $(10.25H^2 + 2H)L + H + 2VH$
    - Embedding layer and LM head: $2VH$
    - Attention mechanism: $2.25H^2$
    - MLP layer: $8H^2$
    - RMSNorm (2 layers): $2H$
    - Additional RMSNorm (last layer): $H$
    - Total per layer: $10.25H^2 + 2H$

2. **LoRA/PiSSA/MiLoRA/LoRA-Dash/KASA**:
  - **Total Parameters**: $11.58HrL$
  - **Proportion**: 0.84%

3. **DoRA**:
  - **Total Parameters**: $(11.58Hr + 5)L$
  - **Proportion**: 0.84%

4. **NEAT**:
  - **Total Parameters**: $(11.58Hr + 10r^2)L$
  - **Proportion**: 0.84%

5. **MoLoRA/GOAT**:
  - **Total Parameters**: $(11.58Hr + 6.66He)L$
    - Attention mechanism: $4.25Hr + 3He$
    - MLP layer: $7.33Hr + 3.66He$
    - Total per layer: $11.58Hr + 6.66He$
  - **Proportion**: 0.96%

6. **HydraLoRA**:
  - **Total Parameters**: $(4.91Hr + 6.66Hr/e + 6.66He)L$
  - **Proportion**: 0.84%

7. **AdaMoLE**:
  - **Total Parameters**: $(11.58Hr + 6.66He + 6.66H)L$
  - **Proportion**: 0.97%

## H.2. FLOPs Analysis

Here, we mainly analyze the forward FLOPs. Since LLaMA 2 7B uses GQA (Grouped Query Attention) and SwiGLU FFN, the calculation of FLOPs differs from that of standard Transformers. Here, we assume that all linear layers in the Transformer block are extended with MoE (Mixture of Experts). We assume $H$ represents the model dimension, $s$ denotes sequence lengths, $d$ denotes each expert rank, $e$ indicates the number of experts, total rank $r = ed$, $L$ indicates the number of layers, $V$ indicates the vocabulary size. *Notice each MAC (Multiply-Accumulate Operations) counts as two FLOPs.*

**FLOPs for FT MoE:**

1. MoE linear for $q$ and $o$: The FLOPs are calculated as $2 \cdot (2BsHe + k \cdot 2BsH^2)$.

2. MoE linear for $k$ and $v$: Since LLaMA 2 7B's GQA reduces the number of heads for $k$ and $v$ to $1/8$ of $q$'s heads, the FLOPs are: $2 \cdot (2BsHe + k \cdot 2BsHH/8)$.

3. The FLOPs for $q \cdot k$ and $score \cdot v$ remain independent of $k$, as we only upcycle the linear projection to $e$ copies. The FLOPs for these operations are $2Bs^2H + 2Bs^2H$.

4. MoE linear for $down$ and $gate$: Since LLaMA 2 7B uses SwiGLU FFN, the FLOPs are: $2 \cdot (2BsHe + k \cdot 2BsH \cdot 8/3H)$.

5. MoE linear for $up$: The FLOPs are: $2Bs \cdot 8/3He + k \cdot 2Bs \cdot 8/3HH$.

Across $L$ layers, including the vocabulary embedding transformation, the total FLOPs are:

$$\text{FLOPs}_{\text{Full FT MoE}} = BL \left( \frac{52}{3}esH + \frac{41}{2}ksH^2 + 4s^2H \right) + 2BsHV \tag{84}$$

**FLOPs for GOAT/MoLoRA/HydraLoRA:**

1. MoE linear for $q$ and $o$: The FLOPs are calculated as $2B \cdot (2sH^2 + 2esH + 2k(sHd + sHd))$.

2. MoE linear for $k$ and $v$: Consider the effect of LLaMA 2 7B's GQA on $k$ and $v$ : $2B \cdot (2sH^2/8 + 2esH + 2k(sHd + sHd/8))$.

3. FLOPs for $q \cdot k$ and $score \cdot v$: The FLOPs for these operations are $2Bs^2H + 2Bs^2h$.

4. MoE linear for $down$ and $gate$: Since LLaMA 2 7B uses SwiGLU FFN, the FLOPs are: $2B \cdot (2sH8/3H + 2esH + 2k \cdot (sHd + sd8/3H))$.

5. MoE linear for $up$: The FLOPs are: $2BsH8/3H + 2Bs8/3He + 2k \cdot (Bs8/3Hd + BsrH)$.

Across $L$ layers, including the vocabulary embedding transformation, the total FLOPs are:

$$\text{FLOPs}_{\text{LoRA-MoE}} = BL \left( \frac{52}{3}esH + \frac{41}{2}sH^2 + 4s^2H + \frac{69}{2}ksHd \right) + 2BsHV \tag{85}$$

$$= BL \left( \frac{52}{3}esH + \frac{41}{2}sH^2 + 4s^2H + \frac{69}{2}\frac{k}{e}sHr \right) + 2BsHV \tag{86}$$

