# OpenReview forum: "Make LoRA Great Again: Boosting LoRA with Adaptive Singular Values and Mixture-of-Experts Optimization Alignment"
_ICML.cc/2025/Conference — ICML 2025 poster_

### Official Review · Reviewer_4qHz · 2025-03-09

**Overall Recommendation:** 4

**Summary:**

The paper proposes the GOAT, an SVD-derived LoRA-MoE finetuning framework. Through SVD decomposition, the authors found that existing LoRA finetuning schemes are insufficient due to restrictive training on only specific pre-selected SVD segments. Based on this finding, the authors proposed employing a LoRA-MoE architecture for finetuning, initializing each LoRA expert with different SVD segments and dynamically selecting segments via MoE during training. To enforce alignment of GOAT with full finetuning, specialized optimization for weight and gradient alignments are designed. Based on experiment results against various baselines on different datasets (tasks), GOAT is demonstrated to yield superior performance in almost all cases.

**Claims And Evidence:**

Yes. The paper claims that:

1) the integration of SVD segments via dynamic selection is necessary to achieve good initialization for LoRA(-MoE) finetuning;
2) specialized optimization for alignment to full finetuning is necessary to improve model performance.

These claims are supported by:

- good preliminary empirical analysis in Fig. 1, which shows the effects of initializing different SVD segments for different scenarios;
- good method design demonstrating the integration of SVD segments into LoRA-MoE architecture and the computation of W_res and s for weight and gradient alignment;
- good experiment performance showing SOTA performance.

Thus, the claims made within the paper are well supported.

**Essential References Not Discussed:**

No. To the knowledge of this reviewer, no essential references are missing.

**Experimental Designs Or Analyses:**

Yes. The experiment designs are considered valid.

- A good variety of image and language tasks are considered, with GOAT outperforming baseline methods in most cases.
- The ablation study, as shown in Table 5, compares GOAT with alternative schemes of initialization on only the principal, minor, and random to validate the claim that SVD-based initialization is necessary.
- Table 7, when taken together with Table 1 and Table 3, shows that GOAT does not introduce excessive parameters, and utilizes similar GPU RAM and training time compared to baseline methods. This demonstrates that the experiment evaluates GOAT against baseline methods fairly.
- (Minor Weakness) For Fig. 7, is load balancing used during training? While load balancing is necessary for most MoE training, it also weakens the conclusion of "validates on the effectiveness of each SVD chunk". This is because balanced workload distribution is enforced by the load balancing loss, rather than implicitly achieved through SVD-based initialization alone.

Aside from minor clarification required from Fig. 7, the experiment and analysis design are valid.

**Methods And Evaluation Criteria:**

Yes.

The proposed method is sensible.

- Employing MoE to facilitate dynamic selection of LoRA related to specific SVD segment is a reasonable design.
- The mathematics for deriving W_res and s for alignment appears sensible.

The benchmark datasets are sensible.

- Since GOAT is task-agnostic, both vision and language tasks are used in the experiment.
- The baseline (compared) algorithms are recent, with most published in 2024.

Thus the overall methods and evaluation are sensible.

**Other Comments Or Suggestions:**

None.

**Other Strengths And Weaknesses:**

The authors have investigated an important problem (improving LoRA-finetuning) and proposed an interesting design with solid analyses and good experiment results. However, some additional weaknesses are noted below.

Weakness (not previously addressed):
- (Weakness) The goal of Section 2.2 (Rethinking Scaling Factor) is not understood. What is the final verdict from the exploration? Specifically, how does this exploration relate to alignment wrt full-finetuning?
- (Minor Weakness) The pseudocode of your algorithm should be presented (if necessary, in the appendix).
- (Minor Weakness) On Figure 3.II, please emphasize that the goal is to find W_res and s. Currently, this is not intuitive without scrutiny over the figure (font too small for W_res and s) and explicit rereading of Section 3.3 (please set the closed-form solutions for W_res and s as labelled equations).

**Questions For Authors:**

For ease of reference, these comments/questions regarding weaknesses in the paper are repeated below. Upon providing satisfactory response, the overall recommendation will be changed to Accept (from Weak Accept).
1) (Minor Weakness, More Information) Lemma 3.5 requires (non-intuitive) knowledge about Leaky-ReLU with negative slope of sqrt(5) resulting in Var(A)=1/(3n). Please cite the source or provide additional proof for this information.
2) (Minor Weakness, Clarify) For Fig. 7, is load balancing used during training? While load balancing is necessary for most MoE training, it also weakens the conclusion of "validates on the effectiveness of each SVD chunk". This is because balanced workload distribution is enforced by the load balancing loss, rather than implicitly achieved through SVD-based initialization alone.
3) (Weakness, Revision) The goal of Section 2.2 (Rethinking Scaling Factor) is not understood. What is the final verdict from the exploration? Specifically, how does this exploration relate to alignment wrt full-finetuning?
4) (Minor Weakness, More Information) The pseudocode of your algorithm should be presented (if necessary, in the appendix).
5) (Minor Weakness, Revision) On Figure 3.II, please emphasize that the goal is to find W_res and s. Currently, this is not intuitive without scrutiny over the figure (font too small for W_res and s) and explicit rereading of Section 3.3 (please set the closed-form solutions for W_res and s as labelled equations).

**Relation To Broader Scientific Literature:**

- The research advances ongoing research on improving the performance of LoRA-finetuned models. The paper focuses particularly on addressing deficiencies in parameter initialization. GOAT outperforms existing initialization techniques, such as PiSSA and KaSA. GOAT also introduces another perspective on weight alignment wrt full finetuning orthogonal to preceding works, such as DoRA and analysis by (Shuttleworth, 2024).
- The integration of LoRA with Mixture-of-Experts expands on an emergent trend of Mixture-of-LoRA frameworks, such as MoLE (Wu, 2024) and MixLoRA (Li, 2024). Moreover, the author provides SVD-based analyses to justify the use of Mixture-of-LoRA frameworks to ameliorate low-rank-related deficiencies found in conventional LoRA finetuning.

References:

Shuttleworth, R., Andreas, J., Torralba, A., & Sharma, P. (2024). Lora vs full fine-tuning: An illusion of equivalence. arXiv preprint arXiv:2410.21228.

Wu, X., Huang, S., & Wei, F. (2024). Mixture of lora experts. arXiv preprint arXiv:2404.13628.

Li, D., Ma, Y., Wang, N., Ye, Z., Cheng, Z., Tang, Y., ... & Tang, M. (2024). Mixlora: Enhancing large language models fine-tuning with lora-based mixture of experts. arXiv preprint arXiv:2404.15159.

PiSSA, KaSA, and DoRA are cited within the reviewed paper.

**Theoretical Claims:**

Yes. The critical mathematical proofs in paper are predominantly focused on finetuning alignment (Section 3.3). They are found in Appendix C. Specifically:

- Lemma 3.1-3.4 appears correct, and are generally intuitive.
- (Minor Weakness) Lemma 3.5 requires (non-intuitive) knowledge about Leaky-ReLU with negative slope of sqrt(5) resulting in Var(A)=1/(3n). Please cite the source or provide additional proof for this information.

Aside from the minor clarification required at Lemma 3.5, the critical proofs appear correct.

---

> ### Author Rebuttal · Authors · 2025-04-01
>
> # Response to Reviewer `4qHz`
>
> > Q1: Lemma 3.5 requires (non-intuitive) knowledge about Leaky-ReLU with negative slope of sqrt(5) resulting in Var(A)=1/(3n). Please cite the source or provide additional proof for this information.
> >
> Thanks for your suggestion. We follow the derivation of the commonly used Kaiming initialization[1] (assuming activation function is Leaky ReLU[2]):
>
> Proof: Leaky ReLU is defined as $f(x)=x \text{ if } x \geq 0, \text{else } f(x) = a·x (a = \sqrt{5})$
> Following Kaiming initialization[1], assume input $x$ is zero-mean with variance $\sigma^2$, symmetrically distributed ($P(x \geq 0) = P(x < 0) = \frac{1}{2}$),
> we can obtain output variance as:
> $$Var(f(x)) = \frac{1}{2}(1 + a^2)\sigma^2=3\sigma^2 \quad (a^2 = 5)$$
> To ensure unit variance per layer ($Var(y)=1$), because $Var(y) = n Var(x)$, where $n$ is the layer width (fan-in).
> $n Var(f(x)) = 1 \implies 3n\sigma^2 = 1 \implies \sigma^2 = \frac{1}{3n}$,
> Thus, weights $A$ should be initialized with $Var(A) = \frac{1}{3n}$.
>
> [1] Delving Deep into Rectifiers: Surpassing Human-Level Performance on ImageNet Classification (ICCV2015)
>
> [2] Empirical Evaluation of Rectified Activations in Convolution Network (2015)
>
> > Q2:For Fig. 7, is load balancing used during training? While load balancing is necessary for most MoE training, it also weakens the conclusion of "validates on the effectiveness of each SVD chunk". This is because balanced workload distribution is enforced by the load balancing loss, rather than implicitly achieved through SVD-based initialization alone.
> >
> Thank you for raising this insightful point. In Fig.7, we use load balancing, but the conclusion holds because:
> - Section 2.1 shows single-LoRA initialization via distinct SVD segments (no load balancing), the Fig.1 revealing their dataset-dependent roles.
> - We further ablate the load-balancing in GOAT (2in8, Cars task), which confirms all experts remain active (see table below), proving each SVD chunk contributes meaningfully.
> |GOAT|f1|f2|f3|f4|f5|f6|f7|f8|
> |-|-|-|-|-|-|-|-|-|
> |w/o loadbalance|0.1043|0.1379|0.1275|0.1094|0.1207|0.1405|0.1259|0.1338|
>
> > Q3: The goal of Section 2.2 (Rethinking Scaling Factor) is not understood. What is the final verdict from the exploration? how does this exploration relate to alignment wrt full-finetuning?
> >
> The goal of Section 2.2 is to establish two key insights: (1) Weight initalization alignment alone is insufficient - gradient alignment is equally crucial  (2) The scaling factor s fundamentally controls gradient dynamics.
>
> Section 2.2 experiments (Figure 2) reveal that even with perfect initalization alignment, common choices (s=2) produce small gradient norms and slow convergence.  Increasing s - particularly in low-rank settings - boosts gradient magnitudes and accelerates training. This leads to Lemma 2.2: scaling factor s directly governs gradient behavior, meaning poor s choices degrade optimization dynamics regardless of weight initialization.
>
> This motivates our core contribution in Theorem 3.2 and Theorem 3.5. Theorem 3.2 establishes that both weight initialization and gradient updates must align for $W_{LoRA}$ to track $W_{FFT}$ based on the first insight. In Theorem 3.5, we chose to derive the optimal s to ensure gradient updates alignment, as the second insight and experiment tells us control s is a practical way to adjust the gradient dynamics. Together, these theorems provide a complete framework where proper scaling factor selection enables LoRA to match full fine-tuning performance through principled optimization dynamics.
>
> > Q4: The algorithm pseudocode should be presented.
> >
> Thank you for your reminder. We incorporate the following pseudocode in the revised paper:
>
> Algorithm: GOAT
> Input: $x$ (input),$n$ (input dim), $\eta, \rho$ (hyperparameter), $E$ (num experts)
> - Set Scaling Factor:  $s = \sqrt{\frac{3n\eta}{r}}$
> - SVD Decomposition: $W_0 = U \Sigma V^\top$
> - Initialization ($\forall i \in [1,E]$):
>     - trainable component: $B_0^i = \sqrt{\frac{1}{s\rho}} U' \Sigma'^{1/2}, \quad A_0^i = \sqrt{\frac{1}{s\rho}} \Sigma'^{1/2} V'^\top$
>     - residual component: $W_{\text{res}}^+ = \frac{s}{E} \sum_{i=1}^E B^i_0 A^i_0, \quad \tilde{W_0} = W_0 - W_{\text{res}}^+$
> - Forward ($\forall i \in [1,E]$):
>     - Compute gating weights: $w^i(x)$
>     - Output: $\tilde{W_0}(x) + \sum_{i=1}^E w^i(x) s B^i_0 A^i_0(x)$
>
> > (Q5) On Figure 3.II, please emphasize that the goal is to find W_res and s. Currently, this is not intuitive without scrutiny over the figure (font too small for W_res and s) and explicit rereading of Section 3.3 (please set the closed-form solutions for W_res and s as labelled equations).
> >
> Thanks for your valuable advice. We will revise our figure to make it clearer.

---

### Official Review · Reviewer_5G79 · 2025-03-11

**Overall Recommendation:** 3

**Summary:**

This paper proposed a PEFT method with SVD-structured MoE and theoretical scaling. It initializes LoRA MoE experts with distinct singular value segments, and derives an optimal weight alignment strategy and scaling scheme to improve both convergence speed and performance. Extensive experiments on 25 tasks validates the effectiveness of the proposed method.

**Claims And Evidence:**

Most of the claims are supported, but there are still a few claims that are not convincing to me:

(1) In Theorem 3.1, the authors claim that ‘we can align LoRA with Full FT’, ‘addresses the performance gap in single LoRA architectures’. Actually, the proposed method is still worse than full finetuning in most of the tasks, as shown in the experiments. In my opinion, this proposed method can only reduce the gap between LoRA and full FT, so the contribution here should be clarified.

(2) In Theorem 3.5, the authors derive the optimal scaling factor from an assumption: there is a fixed learning rate ratio between full tuning vs. LoRA. However, as the learning rate (LR) is actually a hyperparameter, we cannot know the optimal LR for finetuning before we really do multiple runs of full FT, which is not applicable in PEFT setting. If the LR for full FT is arbitrarily selected in practice, the derived scaling value looks less useful. Moreover, in real experiments, we usually use a LR scheduler, and the LR ratio between LoRA and FT are even changing along the training trajectory, posing additional challenges for this assumption.

**Essential References Not Discussed:**

No.

**Experimental Designs Or Analyses:**

Yes, all of those in the experiments section.

**Methods And Evaluation Criteria:**

Yes.

**Other Comments Or Suggestions:**

Typo: L195 ‘segement’, Figure 3 ‘Graident Alignment’.

**Other Strengths And Weaknesses:**

The paper could be improved for better clarity. For example, there is no algorithm description for the whole method, so the whole algorithm is splitted into different sections which hinders the readability.

**Questions For Authors:**

No.

**Relation To Broader Scientific Literature:**

The initialization of weights and scaling methods are novel.

**Theoretical Claims:**

No.

---

> ### Author Rebuttal · Authors · 2025-04-01
>
> # Response to Reviewer `5G79`
>
> > Q1:In Theorem 3.1, the authors claim that ‘we can align LoRA with Full FT’, ‘addresses the performance gap in single LoRA architectures’. Actually, the proposed method is still worse than full finetuning in most of the tasks, as shown in the experiments. In my opinion, this proposed method can only reduce the gap between LoRA and full FT, so the contribution here should be clarified.
> >
>
> Thanks for your suggestion. Due to practical limitations (e.g., low-rank approximation, error accumulation), which may impact theoretical alignment, we will address this in the revision.
>
> > Q2:
> In Theorem 3.5, the authors derive the optimal scaling factor from an assumption: there is a fixed learning rate ratio between full tuning vs. LoRA. However, as the learning rate (LR) is actually a hyperparameter, we cannot know the optimal LR for finetuning before we really do multiple runs of full FT, which is not applicable in PEFT setting. If the LR for full FT is arbitrarily selected in practice, the derived scaling value looks less useful.
> Moreover, in real experiments, we usually use a LR scheduler, and the LR ratio between LoRA and FT are even changing along the training trajectory, posing additional challenges for this assumption
> >
>
> Thanks for your insightful question. To clarify, the learning rate ratio in our framework serves as a tunable hyperparameter, not a fixed constant. This means in practice, rather than arbitrarily selecting a full FT learning rate by multiple exhaustive full FT runs, one can first identify an optimal LR specifically for LoRA (which is computationally more feasible), then tune the ratio hyperparameter to implicitly define the corresponding optimal LR for full FT. In fact, similar assumptions and approaches have been validated and utilized effectively in existing literature [1]
>
> Regarding dynamic LR scheduling, it does not impact the alignment. This is because our theoretical framework is grounded on aligning the weight updates: $\eta_{\text{FFT}} \cdot g_{\text{FFT}} = \eta_{\text{LoRA}} \cdot g_{\text{LoRA}}$. i.e. $\frac{\eta_{\text{LoRA}}}{\eta_{\text{FFT}}} g_{\text{LoRA}} \approx g_{\text{FFT}}$, as long as LoRA and full FT share identical learning rate scheduling patterns at each training iteration, the relative LR ratio $\frac{\eta_{\text{LoRA}}}{\eta_{\text{FFT}}}$ remains consistent.
>
> To empirically substantiate this theoretical robustness, we refer readers to **Table 6**, where we report results across various LR settings. Our approach consistently maintains superior performance, exceeding baseline methods by a clear margin of 1.09-2.56 points, demonstrating its practical resilience and broad applicability.
>
> [1] LoRA-GA: Low-Rank Adaptation with Gradient Approximation (NeurIPS2024)
>
> > Q3:The paper could be improved for better clarity. For example, there is no algorithm description for the whole method, so the whole algorithm is splitted into different sections which hinders the readability.
> >
>
> Thank you for your reminder. We incorporate the following pseudocode in the revised paper:
>
> Algorithm: GOAT
> Input: $x$ (input),$n$ (input dim), $\eta, \rho$ (hyperparameter), $E$ (num experts)
> - Set Scaling Factor:  $s = \sqrt{\frac{3n\eta}{r}}$
> - SVD Decomposition: $W_0 = U \Sigma V^\top$
> - Initialization ($\forall i \in [1,E]$):
>     - trainable component: $B_0^i = \sqrt{\frac{1}{s\rho}} U' \Sigma'^{1/2}, \quad A_0^i = \sqrt{\frac{1}{s\rho}} \Sigma'^{1/2} V'^\top$
>     - residual component: $W_{\text{res}}^+ = \frac{s}{E} \sum_{i=1}^E B^i_0 A^i_0, \quad \tilde{W_0} = W_0 - W_{\text{res}}^+$
> - Forward ($\forall i \in [1,E]$):
>     - Compute gating weights: $w^i(x)$
>     - Output: $\tilde{W_0}(x) + \sum_{i=1}^E w^i(x) s B^i_0 A^i_0(x)$
>
> > Q4:Typo: L195 ‘segement’, Figure 3 ‘Graident Alignment’.
> >
>
> Thanks for your valuable advice. We will carefully revise our paper based on your suggestions.

---

### Official Review · Reviewer_9dgB · 2025-03-12

**Overall Recommendation:** 3

**Summary:**

This paper presents GOAT (Great LoRA Mixture-of-Experts), a novel framework to enhance the LoRA MoE structure for fine-tuning LLMs. GOAT (1) adaptively initializes each expert using different SVD segments to integrate relevant priors from pre-trained models, and (2) derives a theoretical scaling factor that aligns LoRA MoE optimization with Full FT by minimizing gradient misalignment. Experiments across 4 multi-task benchmarks demonstrate GOAT’s superior performance than existing LoRA MoE-based methods, closing the gap with Full FT.

**Claims And Evidence:**

Why does the scaling scheme for gradient alignment with Full FT theoretically improve performance? In other words, the gradient update of Full FT may not always be optimal across all scenarios, as it is influenced by factors such as training data and learning rate. Therefore, aligning with Full FT does not necessarily guarantee the best results.

Moreover, the GOAT+ method in Appendix D, which achieves a more precise alignment with Full FT, appears to perform slightly worse than the GOAT method. This raises concerns about whether strict alignment is indeed beneficial in practice.

**Essential References Not Discussed:**

No.

**Experimental Designs Or Analyses:**

1. Could you provide detailed settings and selection strategies for the coefficient of the load balancing loss used in GOAT and other MoE-based methods? Since this coefficient can significantly impact the final performance, a clearer explanation would be beneficial.
2. Given that GOAT improves convergence speed, is it fair to train all methods for the same carefully selected number of epochs detailed in Appendix E.5? Would it be more appropriate to compare the best performance achieved by each method instead?

**Methods And Evaluation Criteria:**

Yes.

**Other Comments Or Suggestions:**

No.

**Other Strengths And Weaknesses:**

No.

**Questions For Authors:**

No.

**Relation To Broader Scientific Literature:**

Somewhat.

**Theoretical Claims:**

Yes, the paper provides correct theoretical derivations to support the claims.

---

> ### Author Rebuttal · Authors · 2025-04-01
>
> # Response to Reviewer `9dgB`
>
> > Q1: Why does gradient alignment with Full FT improve performance theoretically, given that Full FT's updates aren't always optimal due to data/learning rate dependencies?
> >
> Thanks for your insightful question. First, Full FT outperforms LoRA in most cases, making it a natural alignment target in previous works [1].
> Second, For scenarios where Full FT perform poorly, is because that its unregularized fitting capacity in low-data regimes lead to overfitting on intricate patterns and noise in smaller datasets. (e.g., CoLA, MRPC with 8.5K and 3.6K samples only in Table 4)
> In contrast, our method combines Full FT strengths with three key regularizers to mitigate overfitting issues:
> - Low-Rank Updates: low-rank updates enforce robust feature learning and reduce sensitivity to noise [2].
> - MoE Architecture: Experts specialize in distinct patterns, avoiding over-adaptation to spurious variations.
> - SVD initalization:  Experts are initialized by different pretrained SVD-segmented features, enhancing specialization and mitigating overfitting.
> Thus, our method approximates Full FT’s strong fitting ability while avoiding its pitfalls, achieving comparable or superior performance (e.g., CoLA in Tables 4).
>
> [1] LoRA-GA: Low-Rank Adaptation with Gradient Approximation (NeurIPS2024)
>
> [2] LoRA Learns Less and Forgets Less (TMLR2024)
>
> > Q2: Does the slightly worse performance of GOAT+ in Appendix D, despite its more precise alignment with Full FT, suggest that strict alignment may not be beneficial in practice?
> >
> Sorry for the confusion. GOAT+ is not intended as an improvement over GOAT (not a more precise alignment), but rather as a variant that explores a different assumption.
>
> In GOAT, we assign the same scaling factor to each expert, even though each expert is initialized with a different singular value, leading to varying norms. In contrast, GOAT+ adjusts each expert’s scaling factor in proportion to its singular value, ensuring that the product of the scaling factor and singular value is consistent across all experts. While this adjustment doesn't always improve performance, we found the underlying assumption interesting enough to include it in our appendix.
>
> In the ablation study (Table 5), removing the module responsible for aligning with Full FT degrades performance, demonstrating the effectiveness of strict alignment. We will revise the naming in the paper to avoid any confusion.
>
> > Q3:Could you provide detailed settings and selection strategies for the coefficient of the load balancing loss used in GOAT and other MoE-based methods?
> >
> Thanks for your suggestion. We use top-k routing with k=2 and set the coefficient for the balance loss to 1e-3.
> We attach the load-balancing loss coefficient experiment by activating 2 out of 8 experts on Cars.
>
> |coefficient|GOAT|MoLoRA|HydraLoRA|
> |-|-|-|-|
> |1e-1|49.09|49.02|48.45|
> |1e-2|50.52|49.33|**49.45**|
> |1e-3|**53.50**|**50.83**|48.42|
> |1e-4|51.53|49.03|48.52|
> |0|49.85|48.02|49.06|
>
> We can observe that setting the coefficient too low (e.g., 0 or 1e-4) leads to expert imbalances, which in turn degrades performance. Conversely, excessively high coefficients (e.g., 0.01 or 0.1) can disrupt the normal learning process. Our results show that a coefficient of 1e-3 achieves the best tradeoff in GOAT/MoLoRA between balancing expert load and maintaining stable learning.
>
> Notably, GOAT consistently outperforms across all tested coefficients, demonstrating its robustness in these settings.
>
> > Q4: Given GOAT's faster convergence, is it fair to train all methods for the same number of epochs (Appendix E.5)? Should we compare the best performance of each method instead?
> >
> To clarify, our evaluation strategy indeed follows your suggestion by comparing the best performance achieved by each method. Specifically, we train each model for a sufficiently large number of epochs so that the loss converges to a stable plateau, then evaluate the model at every epoch and select the best-performing result for all baselines.
>
> For the epoch number:
>
> - NLG/NLU: We use more epochs than previous studies to ensure convergence. For example, while prior work often uses just one epoch for NLG tasks [1,2], we employ five epochs to guarantee convergence.
> - Commonsense Reasoning: We strictly follow prior work [3] by using a large dataset (approximately 170K samples) to ensure thorough convergence. We then directly compare our results with the best-reported values from earlier studies, where our method still achieves superior performance.
> - CV: we retain the original epoch settings from previous work [4], as they ensure proper convergence for each task.
>
> [1] KaSA: Knowledge-Aware Singular-Value Adaptation of Large Language Models (ICLR2025) (ICLR2025)
>
> [2] LoRA-GA: Low-Rank Adaptation with Gradient Approximation (NeurIPS2024)
>
> [3] DoRA: Weight-Decomposed Low-Rank Adaptation (ICML2024)
>
> [4] Localizing Task Information for Improved Model Merging and Compression (ICML 2024)

---

### Official Review · Reviewer_ETXs · 2025-03-12

**Overall Recommendation:** 3

**Summary:**

The paper proposes a novel fine-tuning framework for LoRA (Low-Rank Adaptation) MoE (Mixture-of-Experts). Two challenges identified in the paper: 1) how to design an effective initialization for the matrices  A  and  B across different experts. 2)  unaligned optimization leads to large gradient gap and slow convergence rate.
Accordingly, the paper first proposes initializing LoRA MoE experts with distinct singular value segments, allowing the router to select the appropriate prior information. It then derives an optimal weight alignment strategy and a theoretical scaling scheme to improve gradient alignment.
Extensive experiments on 25 tasks demonstrate that the method's superiority while maintaining scalability. Compared with Full fine-tuning, the proposed method shows comparable or even better performance.

### update after rebuttal
My concerns about experiment  evaluations are mostly addressed. Thus, I remain positive about this paper.

**Claims And Evidence:**

The claims are mostly supported by clear and convincing evidence.

**Essential References Not Discussed:**

NA

**Experimental Designs Or Analyses:**

I have checked the experiment part. The performance evluation metrics are not clear in Table 1-4.

**Methods And Evaluation Criteria:**

The proposed method and evaluation criteria are well-aligned with the problem. A better LoRA initialization method is crucial for narrowing the performance gap between parameter-efficient tuning and full fine-tuning. It is commendable that the proposed approach is validated on extensive CV and NLP tasks.

**Other Comments Or Suggestions:**

It would be better to incoporate the practical importance of LoRA MoE in the introduction by highlighting its real-world applications, like multi-task or multi-domain scenarios.

**Other Strengths And Weaknesses:**

Strengths：

1.	The motivation is clear. Relying solely on either the principal or minor singular values is not ideal and does not guarantee optimal performance across different datasets. In LoRA MoE, different experts can be responsible for different parts of the task. Moreover, the effects of the scaling factor are also crucial for optimization

2. Evaluations are quite comprehensive. Significant Improvement over othe LoRA MoE methods, and in NLP benchmarks the proposed method is even better than Full FT. The proposed method also shows good scalability across different rank and different expert numbers.

Weakness:

1. The routing strategy in the proposed method is not clear. In Formula (12), it introduces a soft routing stratey, while in Figure 6, it shows the results of different activation ratio. A more clear description and analysis of the MoE router’s behavior would provide valuable insights, particularly in terms of its latency, and potential routing biases.
2. The paper does not adequately introduce the practical applications of LoRA MoE, nor does it sufficiently demonstrate the real-world impact of the proposed method.
3. To my knowledge, existing approaches typically train multiple LoRA experts for different tasks. However, this paper does not report results on multi-domain datasets, especially in Image classification and NLU tasks, which limits its practical relevance.

**Questions For Authors:**

1. In table 1, the single LoRA method comparison part, methods like PiSSA and MiLoRA achieves worse performance than LoRA. Are there any analysis on this phenomenon?
2.  Do you experiment with alternative routing techniques in LoRA MoE? Could you discuss how these different strategies impact performance?

**Relation To Broader Scientific Literature:**

The intialization of AB matrices in LoRA is a rising topic and has been studied in previous methods, including PISSA [1] (tuning the principal components) and MiLoRA [2] (tuning minor singular components). For LoRA-MoE, the initialization has not been well studied, the proposed method divides the SVD of W into different segments and allocate segments to different experts. From this perspective, the proposed method is a timely study.

[1] PiSSA: Principal Singular Values and Singular Vectors Adaptation of Large Language Models
[2] Milora (MiLoRA): Harnessing Minor Singular Components for Parameter-Efficient LLM Finetuning

**Theoretical Claims:**

Yes, I have checked the theorems in Section 3.3. Theoretical Optimization Alignment, both the initialization and gradient alignment. The claims seem reasonable.

---

> ### Author Rebuttal · Authors · 2025-04-01
>
> # Response to Reviewer `ETXs`
>
> > Q1: The performance evaluation metrics are unclear.
> >
>
> Sorry for the confusion. Here is a more detailed explanation of our performance metrics:
>
> - NLU & CV: Accuracy, except for CoLA (Matthew’s correlation). See Appendix E.1 for details.
> - CommonSense: Exact match.
> - NLG: GSM8k(Exact match), Human-Eval(Pass@) and mtbench: (First-turn score (GPT-4))
>
> We will incorporate these clarifications into our revised version of the paper.
>
> > Q2: Clearer description and analysis of the MoE router’s strategy and behavior, especially regarding latency and potential biases, would be valuable.
> >
>
> To clarify, we use a top-k routing strategy (Eq(10) and (12)), where each token selects the top k experts based on the highest router logits.
>
> We offer insights about topk hyperparameter, routing biases and latency in Figure 6, 7, and Table 7:
>
> - **Top‑k Hyperparameter:** Figure 6 shows performance vs. the k/E (E is total number of experts) activation ratio in top-k routing. Activating 2 out of 8 experts balances sparsity and performance, so we use this setting in Tables 1–4.
> - **Routing Biases and Load Balance**: Figure 7 shows token distribution across experts. CV and NLU tasks exhibit balanced expert usage, while NLG tasks favor the first two experts, suggesting larger SVD chunks play a key role in complex generation, aligning with PiSSA’s insights.
> - **Latency:** Section 4.9 (Computation Analysis) and Appendix F.1 provide a detailed breakdown of latency and computational efficiency.
>
> > Q3: The paper lacks an adequate introduction to the practical applications and real-world impact of LoRA MoE.
> >
>
> Thanks for your advice. We will incorporate additional discussion on the practical applications and real-world impact of LoRA MoE into the revised version of the paper.
>
> MoE is popular for managing large parameters while activating only a sparse subset during inference, making it ideal for large-scale models. However, in Section 4.9 and Table 7, without optimization, fully fine-tuning an MoE model significantly increases trainable parameters and FLOPs compared to Full FT.
>
> LoRA MoE addresses these challenges by replacing experts with low-rank matrices, reducing computation, preserving MoE benefits, and enabling faster training, lower memory usage, and reduced energy consumption—crucial for resource-limited or real-time applications.
>
> For instance, in NLP, where large-scale models are common, LoRA MoE achieves SOTA performance with lower computational cost. This efficiency benefits industries like autonomous driving, healthcare[1], where lower latency and costs enhance performance and scalability.
>
> Overall, LoRA MoE balances MoE's model capacity with cost-effective deployment, making it adaptable to various real-world applications.
>
> [1]Hydralora: An asymmetric lora architecture for efficient fine-tuning(NeurIPS2024)
>
> > Q4:This paper doesn’t report results on multi-domain datasets.
> >
>
> We actually conducted experiments on Commonsense using a multi-domain setting. In Table 3, our evaluation method for commonsense reasoning follows a classic multi-domain setting, following prior work[1]. We train on a 170K multi-task mixed dataset and evaluate on **8 datasets**. Our approach outperforms the single LoRA method by at least 1.2 points and the LoRA MoE method by at least 1.6 points.
>
> [1] DoRA: Weight-Decomposed Low-Rank Adaptation(ICML2024)
>
> > Q5:An analysis of why PiSSA and MiLoRA perform worse than LoRA in Table 1.
> >
>
> To clarify, previous works[1,2] show that PiSSA and MiLoRA don't always outperform LoRA. KaSA found that PiSSA accelerates convergence but uses limited pre-trained knowledge at lower ranks, limiting performance. Similarly, MiLoRA’s minimal adjustments to pre-trained weights often fail to improve over LoRA.  In Table 1, we adopt the same rank settings as KaSA and reach the same conclusion.
>
> In contrast, our method consistently achieves superior performance across both low and high ranks by effectively balancing convergence speed and final performance, as demonstrated in Tables 1–4 and Figure 5.
>
> [1]MiLoRA: Harnessing Minor Singular Components for Parameter-Efficient LLM Finetuning (NAACL2025)
> [2]KaSA: Knowledge-Aware Singular-Value Adaptation of Large Language Models (ICLR2025)
>
> > Q6. Do you explore alternative routing techniques in LoRA MoE, and how do they affect performance?
> >
>
> Thanks for your suggestion. In our paper, we use the commonly adopted top-k routing and set k=2 based on the analysis in Figure 6. we primarily adopt top‑k routing. We extend our experiments to include alternative routing strategies such as top-p routing and a top‑k variant with shared experts below.
>
> | Routing Strategy | avg.  ACC |
> | --- | --- |
> | **Ours(top-k=2)** | **81.30** |
> | top-p(0.25) | 79.40 |
> | top-k + share expert | 78.68 |
>
> We find that, compared to other approaches, setting k=2 achieves the best performance.We will incorporate these into our revised version of the paper.

---

### Decision · Program_Chairs · 2025-05-01

**Decision:**

Accept (poster)

**Comment:**

The paper presents GOAT, a fine-tuning framework based on LoRA for MoE architectures. The key idea is to initialize the experts' weights using different pieces of an SVD and allowing a router to dynamically select the right expert during training. There's also some theoretical justification showing an equivalence to full fine-tuning of the MoE. The paper is generally well written with solid experimental evidence. Concerns were raised regarding the presentation (particularly, the fact that the method is not clearly written down in terms of pseudocode) and some ambiguities around the choice of scaling factors, but the authors gave a satisfactory response. Overall, I recommend an accept.